# OTTC: A DIFFERENTIABLE ALIGNMENT APPROACH TO AUTOMATIC SPEECH RECOGNITION

## ABSTRACT

The Connectionist Temporal Classification (CTC) and transducer-based models are widely used for end-to-end (E2E) automatic speech recognition (ASR). These methods maximize the marginal probability over all valid alignments within the probability lattice over the vocabulary during training. However, research has shown that most alignments are highly improbable, with the model often concentrating on a limited set, undermining the purpose of considering all possible alignments. In this paper, we propose a novel differentiable alignment framework based on a one-dimensional optimal transport formulation, enabling the model to learn a single alignment and perform ASR in an E2E manner. We define a pseudo-metric, called Sequence Optimal Transport Distance (SOTD), over the sequence space and highlight its theoretical properties. Based on the SOTD, we propose Optimal Temporal Transport Classification (OTTC) loss for ASR and contrast its behavior with that of CTC. Experimental results on the English Librispeech and AMI datasets demonstrate that our method achieves competitive performance compared to CTC in ASR. We believe this work opens up a potential new direction for research in ASR, offering a foundation for the community to further explore and build upon.

## 1 INTRODUCTION

In the literature, two primary approaches to automatic speech recognition (ASR) have emerged, i.e., hybrid systems and end-to-end (E2E) models. In hybrid approaches, a deep neural network-hidden Markov model (DNN-HMM) (Morgan & Bourlard, 1990; Bourlard & Morgan, 2012; Young, 1996; Povey, 2005; Abdel-Hamid et al., 2012; Graves et al., 2013a; Dahl et al., 2012) system is typically trained, where the DNN is optimized by minimizing cross-entropy loss on the forced alignments generated for each frame of audio embeddings from a hidden Markov model-Gaussian mixture model (HMM-GMM). One notable disadvantage of the hybrid approach is that the model cannot be optimized in an E2E manner, which may result in suboptimal performance (Hannun, 2014). More recently, E2E models for ASR have become very popular due to their superior performance. There are three popular approaches for training an E2E model: (i) attention-based encoder-decoder (AED) models (Chan et al., 2015; Radford et al., 2023; Watanabe et al., 2017; Prabhavalkar et al., 2023), (ii) using Connectionist Temporal Classification (CTC) loss (Graves et al., 2006; Graves & Jaitly, 2014), and (iii) neural Transducer-based models (Graves, 2012; Kuang et al., 2022; Graves et al., 2013b). AED models use an encoder to convert the input audio sequence into a hidden representation. The decoder, which is typically auto-regressive, generates the output text sequence by attending to specific parts of the input through an attention mechanism, often referred to as soft alignment (Yan et al., 2022) between the audio and text sequences. This design, however, can make it challenging to obtain word-level timestamps and to do teacher-student training with soft labels. Training AED models also requires a comparatively large amount of data, which can be prohibitive in low-resource setups. In contrast to AED models, CTC and transducer-based models maximize the marginal probability of the correct sequence of tokens (transcript) over all possible valid alignments (paths), often referred to as hard alignment (Yan et al., 2022). However, recent research has shown that only a few paths, which are dominated by blank labels, contribute meaningfully to the marginalization, leading to the well-known peaky behavior that can result in suboptimal ASR performance (Zeyer et al., 2021). Unfortunately, it is not possible to directly identify these prominent paths, or those that do

not disproportionately favor blank labels, in advance within E2E models. This observation serves as the main motivation of our work.

In this paper, we introduce the Optimal Temporal Transport Classification (OTTC) loss function, a novel approach to ASR where our model jointly learns temporal sequence alignment and audio frame classification. OTTC is derived from the Sequence Optimal Transport Distance (SOTD) framework, which is also introduced in this paper and defines a pseudo-metric for finite-length sequences. At the core of this framework is a novel, parameterized, and differentiable alignment model based on one-dimensional optimal transport, offering both simplicity and efficiency, with linear time and space complexity relative to the largest sequence size. This design allows OTTC to be fast and scalable, maximizing the probability of exactly one path, which, as we demonstrate, helps avoid the peaky behavior commonly seen in CTC based models.

To summarize, our contributions are the following:

1. We propose a novel, parameterized, and differentiable sequence-to-sequence alignment model with linear complexity both in time and space.

2. We introduce a new framework, Sequence Optimal Transport Distance (SOTD), to compare finite-length sequences, examining its theoretical properties and providing guarantees on the existence and characteristics of a minimum.

3. We derive a new loss function, Optimal Temporal Transport Classification (OTTC), specifically designed for Automatic Speech Recognition (ASR) tasks.

4. Finally, we conduct proof-of-concept experiments on the English Librispeech (Panayotov et al., 2015) and AMI (Carletta et al., 2005) datasets, demonstrating that our method achieves promising performance in E2E ASR while addressing the peaky behavior issues.

## 2 RELATED WORK

**CTC loss.** The CTC criterion (Graves et al., 2006) is a versatile method for learning alignments between sequences. This versatility has led to its application across various sequence-to-sequence (seq2seq) tasks (Liu et al., 2020; Chuang et al., 2021; Yan et al., 2022; Gu & Kong, 2021; Graves & Schmidhuber, 2008; Molchanov et al., 2016). However, despite its widespread use, CTC has numerous limitations that impact its effectiveness in real-world applications. To address issues such as peaky behavior (Zeyer et al., 2021), label delay (Tian et al., 2023), and alignment drift (Sak et al., 2015), researchers have proposed various extensions. These extensions aim to refine the alignment process, ensuring better performance across diverse tasks. Delay-penalized CTC (Yao et al., 2023) and blank symbol regularization (Yang et al., 2023; Zhao & Bell, 2022; Bluche et al., 2015) attempt to mitigate label delay issues. Other works have tried to control alignment through teacher model spikes (Ghorbani et al., 2018; Kurata & Audhkhasi, 2019) or external supervision (Zeyer et al., 2020; Senior et al., 2015; Plantinga & Fosler-Lussier, 2019), though this increases complexity. Recent advancements like Bayes Risk CTC offer customizable, end-to-end approaches to improve alignment without relying on external supervision (Tian et al., 2023).

**Transducer loss.** The transducer loss was introduced to address the conditional independence assumption of CTC by incorporating a predictor network (Graves, 2012). However, similarly to CTC, transducer models suffer from label delay and peaky behavior (Yu et al., 2021). To mitigate these issues, several methods have been proposed, such as e.g., Pruned RNN-T (Kuang et al., 2022) which prunes alignment paths before loss computation, FastEmit (Yu et al., 2021) which encourages faster symbol emission, delay-penalized transducers (Kang et al., 2023) which add a constant delay to all non-blank log-probabilities, and minimum latency training (Shinohara & Watanabe, 2022) which augments the transducer loss with the expected latency. Further extensions include CIFTransducer (CIF-T) for efficient alignment (Zhang et al., 2024), self-alignment techniques (Kim et al., 2021), and lightweight transducer models using CTC forced alignments (Wan et al., 2024).

Over the years, the CTC and transducer-based ASR models have achieved state-of-the-art performance. Despite numerous efforts to control alignments and apply path pruning, the fundamental formulation of marginalizing over all valid paths remains unchanged and directly or indirectly contributes to several of the aforementioned limitations. Instead of marginalizing over all valid paths

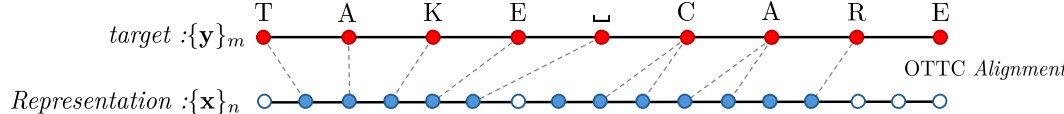

Figure 1: *Example of an alignment between embeddings of frames and target sequence.* The red bullets represent the elements of the target sequence $\{y\}_m$, while the blue bullets indicate the frame embeddings $\{x\}_n$. In OTTC, the alignment guides the prediction model $F$ in determining which frames should map to which labels. Additionally, the alignment model has the flexibility to leave some frames unaligned, as represented by the blue-and-white bullets, allowing those frames to be dropped during inference.

as in CTC and transducer models, we propose a differential alignment framework based on optimal transport which can jointly learn a single alignment and perform ASR task in an E2E manner.

## 3 PROBLEM FORMULATION

We define $\mathcal{U}_{\leq N}^d = \bigcup_{n \leq N} \mathcal{U}_n^d$ to be the set of all $d$-dimensional vector sequences of length at most $N$. Let us consider a distribution $\mathcal{D}_{\mathcal{U}_{\leq N}^d \times \mathcal{U}_{\leq N}^d}$ and pairs of sequences $(\{x_i\}_{i=1}^n, \{y_i\}_{i=1}^m)$ of length $n$ and $m$ drawn from $\mathcal{D}_{\mathcal{U}_{\leq N}^d \times \mathcal{U}_{\leq N}^d}$. For notational simplicity, the sequences of the pairs $(\{x_i\}_{i=1}^n, \{y_i\}_{i=1}^m)$ will be respectively denoted by $\{x\}_n$ and $\{y\}_m$ in the following. The goal in seq2seq tasks is to train a classifier that can accurately predict the target sequence $\{y\}_m$ from the input sequence $\{x\}_n$, enabling it to generalize to unseen examples. Typically, $n \neq m$, creating challenges for accurate prediction as there is no natural alignment between the two sequences. In this paper, we introduce a framework to address this class of problems, applying it specifically to the ASR domain. In this context, the first sequence $\{x\}_n$ represents an audio signal, where each vector $x_i \in \mathbb{R}^d$ corresponds to a time frame in the acoustic embedding space. The second sequence $\{y\}_m$ is the textual transcription of the audio, where each element $y_i$ belongs to a predefined vocabulary $L = \{l_1, \ldots, l_{|L|}\}$, such that $\{y\}_m \in L^m$, where $L^m$ denotes the set of all $m$-length sequences formed from the vocabulary $L$.

## 4 OPTIMAL TEMPORAL TRANSPORT CLASSIFICATION (OTTC)

The core idea is to model the alignment between two sequences as a mapping to be learned along with the frame labels (see Figure 1). Actually, as the classification of audio frames improves, inferring the correct alignment becomes easier. Conversely, accurate alignments also improve frame classification. This mutual reinforcement between alignment and classification highlights the benefit of addressing both tasks simultaneously, contrasting with traditional hybrid models that treat them as separate tasks (Morgan & Bourlard, 1990). To achieve this, we propose the Sequence Optimal Transport Distance (SOTD), a framework for constructing pseudo-metrics over the sequence space $\mathcal{U}_{\leq N}^d$, based on a differentiable, parameterized model that learns to align sequences. Using this framework, we derive the Optimal Temporal Transport Classification (OTTC) loss, which allows the model to learn both the alignment and the classification in a unified manner.

**Notation.** In the following we will denote $[\![1, n]\!] = \{1, \ldots, n\}$.

### 4.1 PRELIMINARIES

**Definition 1.** *Discrete monotonic alignment. Given two sequences $\{x\}_n$ and $\{y\}_m$, and a set of index pairs $\mathbf{A} \subset [\![1, n]\!] \times [\![1, m]\!]$ representing their alignment, we say that $\mathbf{A}$ is a discrete monotonic alignment between the two sequences if:*

- **Complete alignment of $\{y\}_m$:** Every element of $\{y\}_m$ is aligned, i.e.,

$$\forall j \in [\![1, m]\!], \exists k \in [\![1, n]\!], \ (k, j) \in \mathbf{A}.$$

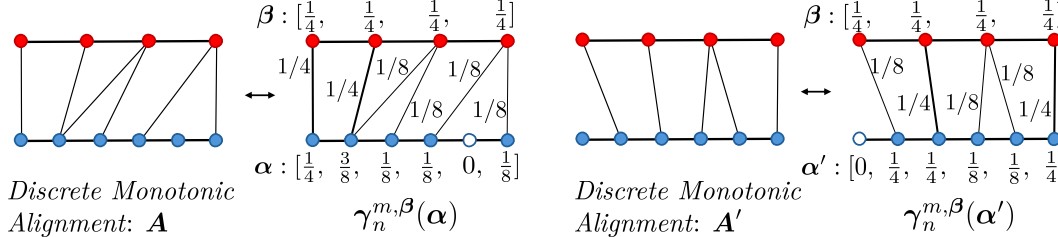

Figure 2: ***Discrete monotonic alignment as 1D OT solution.*** A discrete monotonic alignment represents a temporal alignment between two sequences (target on top, frame embeddings on bottom). It can be modeled by $\gamma_n^{m,\boldsymbol{\beta}}$, as illustrated in the graph. The thickness of the links reflects the amount of mass $\gamma_n^{m,\boldsymbol{\beta}}(\boldsymbol{\alpha})_{i,j}$ transported, with thicker links corresponding to higher mass.

- **Monotonicity:** The alignment is monotonic, meaning that for all $(i, j), (k, l) \in \mathbf{A}$

$$i \leq k \;\Rightarrow\; j \leq l.$$

Discrete monotonic alignments model the relationship between temporal sequences, such as those in ASR, by determining which frame should predict which target. The conditions imposed on the target sequence $\{\mathbf{y}\}_m$ ensure that no target element is omitted, while the absence of similar constraints on the source sequence $\{x\}_n$ allows certain audio frames to be considered irrelevant and dropped (see Figure 2). The monotonicity condition preserves the temporal order, ensuring the sequential structure is maintained. In the following sections, we will develop a model capable of differentiating within the space of discrete monotonic alignments.

### 4.2 DIFFERENTIABLE TEMPORAL ALIGNMENT WITH OPTIMAL TRANSPORT

In the following, we introduce 1D OT and define our alignment model. Consider the 1D discrete distributions $\mu[\boldsymbol{\alpha}, n]$ and $\nu[\boldsymbol{\beta}, m]$, expressed as superpositions of $\delta$ measures, a distribution that is zero everywhere except at a single point, where it integrates to 1 :

$$\mu[\boldsymbol{\alpha}, n] = \sum_{i=1}^{n} \alpha_i \delta_i \quad \text{and} \quad \nu[\boldsymbol{\beta}, m] = \sum_{i=1}^{m} \beta_i \delta_i. \tag{1}$$

The bins of $\mu[\boldsymbol{\alpha}, n]$ and $\nu[\boldsymbol{\beta}, m]$ are $[\![1, n]\!]$ and $[\![1, m]\!]$, respectively, whereas the weights $\alpha_i$ and $\beta_i$ are components of the vectors $\boldsymbol{\alpha} \in \Delta^n$ and $\boldsymbol{\beta} \in \Delta^m$, with $\Delta^n$ the simplex set defined as $\Delta^n = \{\mathbf{v} \in \mathbb{R}^n | 0 \leq v_i \leq 1, \sum_{i=1}^{n} v_i = 1\} \subset \mathbb{R}^n$. Optimal transport theory provides an elegant and versatile framework for computing distances between distributions such as $\mu[\boldsymbol{\alpha}, n]$ and $\nu[\boldsymbol{\beta}, m]$, depending on the choice of the cost function (Peyré & Cuturi, 2019) (chapter 2.4). One such distance is the 2-Wasserstein distance $\mathcal{W}_2$, which measures the minimal cost of transporting the weight of one distribution to match the other. This distance is defined as

$$\mathcal{W}_2(\mu[\boldsymbol{\alpha}, n], \nu[\boldsymbol{\beta}, m]) = \min_{\boldsymbol{\gamma} \in \Gamma^{\boldsymbol{\alpha}, \boldsymbol{\beta}}} \sum_{i,j=1}^{n,m} \gamma_{i,j} \|i - j\|_2^2, \tag{2}$$

were $\|i - j\|_2^2$ is the cost of moving weight from bin $i$ to bin $j$ and $\gamma_{i,j}$ is the amount of mass moved from $i$ to $j$. The optimal coupling matrix $\boldsymbol{\gamma}^*$ is searched within the set of valid couplings $\Gamma^{\boldsymbol{\alpha}, \boldsymbol{\beta}}$ defined as

$$\Gamma^{\boldsymbol{\alpha}, \boldsymbol{\beta}} = \{\boldsymbol{\gamma} \in \mathbb{R}_+^{n \times m} | \boldsymbol{\gamma} \mathbf{1}_m = \boldsymbol{\alpha} \text{ and } \boldsymbol{\gamma}^T \mathbf{1}_n = \boldsymbol{\beta}\}. \tag{3}$$

This constraint ensures that the coupling conserves mass, accurately redistributing all weights between the bins. A key property of optimal transport in 1D is its monotonicity (Peyré, 2019). Specifically, if there is mass transfer between bins $i$ and $j$ (i.e., $\gamma_{i,j}^* > 0$) and similarly between bins $k$ and $l$ (i.e., $\gamma_{k,l}^* > 0$), then it must hold that $i \leq k \Rightarrow j \leq l$. Consequently, when $\boldsymbol{\beta}$ has no zero components —meaning every bin from $\nu$ is reached by the transport— the set $\{(i, j) \in [\![1, n]\!] \times [\![1, m]\!] \mid \boldsymbol{\gamma}_{i,j}^* > 0\}$

satisfies the conditions of Definition 1, thereby forming a discrete monotonic alignment. This demonstrates that the optimal coupling can effectively model such alignments (see Figure 2).

**Note:** In the 1D case, the solution $\boldsymbol{\gamma}^*$ is unique and depends only on the number of distinct bins and their weights, not their specific values. Thus, the choice of $[\![1, n]\!]$ and $[\![1, m]\!]$ as bins is arbitrary (Peyré, 2019).

**Parameterized and differentiable temporal alignment.** Given any sequences length $n$ and $m$ and $\boldsymbol{\beta}$ with no zero components, we can define the alignment function $\boldsymbol{\gamma}_n^{m,\boldsymbol{\beta}}$

$$\boldsymbol{\gamma}_n^{m,\boldsymbol{\beta}} : \mathbb{R}^n \to \Gamma^{*,\boldsymbol{\beta}}[n] \tag{4}$$

$$\boldsymbol{\alpha} \mapsto \boldsymbol{\gamma}^* = \arg\min_{\boldsymbol{\gamma} \in \Gamma} \mathcal{W}(\mu[\boldsymbol{\alpha}, n], \nu[\boldsymbol{\beta}, m]), \tag{5}$$

where $\Gamma^{*,\boldsymbol{\beta}}[n]$ is the space of all 1D transport solutions between $\mu[\boldsymbol{\alpha}, n]$ and $\nu[\boldsymbol{\beta}, m]$ for any $\boldsymbol{\alpha}$. Differently from $\boldsymbol{\beta}$, $\boldsymbol{\alpha}$ may have zero components, giving the model the flexibility to suppress certain bins, which acts similarly to a blank token in traditional models. In the context of ASR, $\boldsymbol{\alpha}$ and $\boldsymbol{\beta}$ can be termed as OT weights and label weights, respectively.

**Lemma 1:** *The function $\boldsymbol{\alpha} \mapsto \boldsymbol{\gamma}_n^{m,\boldsymbol{\beta}}$ is bijective from $\mathbb{R}^n$ to $\Gamma^{*,\boldsymbol{\beta}}[n]$ .*

*Proof.* The proof can be found on Appendix A.2.1.

**Proposition 1**. *Discrete Monotonic Alignment Approximation Equivalence. For any $\boldsymbol{\beta}$ that satisfies the condition above, any discrete set of alignments $\boldsymbol{A} \subset [\![1, n]\!] \times [\![1, m]\!]$ between sequences of lengths $n$ and $m$ can be modeled by $\boldsymbol{\gamma}_n^{m,\boldsymbol{\beta}}$ through the appropriate selection of $\boldsymbol{\alpha}$, i.e.,*

$$\forall \boldsymbol{A}, \exists \boldsymbol{\alpha} \in \Delta^n, (i, j) \in \boldsymbol{A} \iff \boldsymbol{\gamma}_n^{m,\boldsymbol{\beta}}(\boldsymbol{\alpha})_{i,j} > 0. \tag{6}$$

*Proof.* The proof can be found on Appendix A.2.2.

Thus, we have defined a family of alignment functions $\boldsymbol{\gamma}_n^{m,\boldsymbol{\beta}}$ that are capable of modeling any discrete monotonic alignment, which can be chosen or adapted based on the specific task at hand. The computational cost of these alignment functions is low, as the bins are already sorted, eliminating the need for additional sorting. This results in linear complexity $O(\max(n, m))$ depending on the length of the longest sequence (see Algorithm A.1.1 in the Appendix). Furthermore, these alignments are differentiable, with $\boldsymbol{\gamma}_n^{m,\boldsymbol{\beta}}(\boldsymbol{\alpha})_{i,j}$ explicitly expressed in terms of $\boldsymbol{\alpha}$ and $\boldsymbol{\beta}$, allowing direct computation of the derivative $\frac{d\boldsymbol{\gamma}_n^{m,\boldsymbol{\beta}}(\boldsymbol{\alpha})_{i,j}}{d\boldsymbol{\alpha}}$ via its analytical form.

### 4.2.1 SEQUENCES-TO-SEQUENCES DISTANCE

In this section, we will use the previously designed alignment functions to build a pseudo-metric over sets of sequences $\mathcal{U}_{\leq N}^d$.

**Definition 1.** *Sequences Optimal Transport Distance (SOTD). Consider an $n$-length sequence $\{\boldsymbol{x}\}_n \in \mathcal{U}_{\leq N}^d$, an $m$-length sequence $\{\boldsymbol{y}\}_m \in \mathcal{U}_{\leq N}^d$, $p = \max(n, m)$, and $q = \min(n, m)$. Let $C : \mathbb{R}^d \times \mathbb{R}^d \to \mathbb{R}_+$, be a differentiable positive cost function. Considering $r \in \mathbb{N}^*$ and a family of vectors $\{\boldsymbol{\beta}\}_N = \{\boldsymbol{\beta}_1 \in \mathbb{R}, \boldsymbol{\beta}_2 \in \mathbb{R}^2, \ldots, \boldsymbol{\beta}_N \in \mathbb{R}^N\}$ with no zero components, we define the SOTD $\mathcal{S}_r(\{\boldsymbol{x}\}_n, \{\boldsymbol{y}\}_m)$ as*

$$\mathcal{S}_r(\{\boldsymbol{x}\}_n, \{\boldsymbol{y}\}_m) = \min_{\boldsymbol{\alpha} \in \Delta^n} \Big( \sum_{i,j=1}^{n,m} \boldsymbol{\gamma}_p^{q,\boldsymbol{\beta}_q}(\boldsymbol{\alpha})_{i,j} \cdot C(\boldsymbol{x}_i, \boldsymbol{y}_j)^r \Big)^{1/r}. \tag{7}$$

Note that $\boldsymbol{\beta}_q$ obviously depends on $q$, but could a priori depend on $\{\boldsymbol{x}\}_n$ and $\{\boldsymbol{y}\}_m$. To simplify the notation, we will only denote its dependence on $q$. However, all the results in this section remain valid under such dependencies, as long as $\boldsymbol{\beta}_q$ components never becomes zero.

**Proposition 2.** *Validity of the definition. SOTD is well-defined, meaning that a solution to the problem always exists, although it may not be unique.*

*Proof.* The proof and the discussion about the non-unicity is conducted in Appendix A.2.3.

**Proposition 3.** *SOTD is a Pseudo-Metric. If the cost matrix $C$ is a metric on $\mathbb{R}^d$, then $\mathcal{S}_r$ defines a pseudo-metric over the space sequences with at most $N$ elements $\mathcal{U}_{\leq N}^d$.*

*Proof.* The proof can be found in Appendix A.2.4.

Since $\mathcal{S}_r$ is a pseudo-metric, there are sequences $\{\boldsymbol{x}\}_n \neq \{\boldsymbol{y}\}_m$ such that $\mathcal{S}_r(\{\boldsymbol{x}\}_n, \{\boldsymbol{y}\}_m) = 0$. The following proposition describes the conditions under which this occurs.

**Proposition 4.** *Non-Separation Condition. Let $\mathcal{A}$ be the sequence aggregation operator which removes consecutive duplicates, i.e., $\mathcal{A}(\{\dots, \boldsymbol{x}, \boldsymbol{x}, \dots\}) = \{\dots, \boldsymbol{x}, \dots\}$. Let $\mathcal{P}_{\boldsymbol{\alpha}}$ be the sequence pruning operator which removes any element $\boldsymbol{x}_i$ from sequences corresponding to an $\alpha_i = 0$, i.e., $\mathcal{P}_\alpha(\{\dots, \boldsymbol{x}_{i-1}, \boldsymbol{x}_i, \boldsymbol{x}_{i+1}, \dots\}) = \{\dots, \boldsymbol{x}_{i-1}, \boldsymbol{x}_{i+1}, \dots\}$ iff $\alpha_i = 0$. Further, let us consider $\{\boldsymbol{x}\}_n$ and $\{\boldsymbol{y}\}_m$ such that $\{\boldsymbol{x}\}_n \neq \{\boldsymbol{y}\}_m$. Without loss of generality, we assume that $n \geq m$. Then*

$$\mathcal{S}_r(\{\boldsymbol{x}\}_n, \{\boldsymbol{y}\}_m) = 0 \text{ iff } \mathcal{A}(\mathcal{P}_{\alpha^*}(\{\boldsymbol{x}\}_n)) = \mathcal{A}(\{\boldsymbol{y}\}_m), \tag{8}$$

*where $\boldsymbol{\alpha}^*$ is a minimum for which $\mathcal{S}_r(\{\boldsymbol{x}\}_n, \{\boldsymbol{y}\}_m) = 0$. It should be noted that this condition holds also when $C$ is neither symmetric nor satisfies the triangular inequality, but is separated (like the cross-entropy $C_e$ for example).*

*Proof.* See Appendix A.2.5.

The consequence of the previous proposition is that we can learn a transformation through gradient descent using a trainable network $F$ which maps input sequences $\{\boldsymbol{x}\}_n$ to target sequences $\{\boldsymbol{y}\}_m$ (with $n \geq m$) by solving the optimization problem

$$\min_F \mathcal{S}_r(F(\{\boldsymbol{x}\}_n), \{\boldsymbol{y}\}_m) = \min_{F, \boldsymbol{\alpha} \in \Delta^n} \Big( \sum_{i,j=1}^{n,m} \boldsymbol{\gamma}_p^{q, \boldsymbol{\beta}_q}(\boldsymbol{\alpha})_{i,j} \cdot C(F(\{\boldsymbol{x}\}_n)_i, \boldsymbol{y}_j)^r \Big)^{1/r}. \tag{9}$$

We are then guaranteed that a solution $F^*\{\boldsymbol{x}\}_n$ allows us to recover the sequence $\mathcal{A}(\{\boldsymbol{y}\}_m)$. In cases where retrieving repeated elements in $\{\boldsymbol{y}\}_m$ (e.g., double letters) is important, we can intersperse blank labels $\phi \notin L$ between repeated labels as follows: $\{\boldsymbol{y}\}_m = \{\dots, l_i, l_i, \dots\} \rightarrow \{\dots, l_i, \phi, l_i, \dots\}$.

**Note on Dynamic Time Warping (DTW):** It is important to highlight the distinction between our approach and DTW-based (Itakura, 1975) alignment methods, particularly the differentiable variations such as soft-DTW (Cuturi & Blondel, 2018). These methods generally have quadratic complexity (Cuturi & Blondel, 2018), making them significantly more computationally expensive than ours. Furthermore, in DTW-based methods, the alignment emerges as a consequence of the sequences themselves. When the function $F$ is powerful, the model can collapse by generating a sequence $F(\{\boldsymbol{x}\}_n)$ that induces a trivial alignment Haresh et al. (2021). To mitigate this issue, regularization losses (Haresh et al., 2021; Meghanani & Hain, 2024) or constraints on the capacity of $F$ (Vayer et al., 2022; Zhou & la Torre, 2009) are commonly introduced. However, using regularization losses lacks theoretical guarantees and introduces additional hyperparameters, while constraining the capacity of $F$, although more theoretically sound, makes tasks requiring powerful encoders on large datasets impractical. In contrast, our method decouples the computation of the alignment from the transformation function $F$, offering more flexibility to the model as well as built-in temporal alignment constraints and theoretical guarantees against collapse.

### 4.3 APPLICATION TO ASR: OTTC LOSS

In ASR, the target sequences $\{\boldsymbol{y}\}_m$ are $d$-dimensional one-hot encodings of elements from the set $L \cup \{\phi\}$, where $\phi$ is a blank label used to separate repeated labels. The encoder $F$ predicts the label probabilities for each audio frame, such that

$$F(\{\boldsymbol{x}\}_n) = \{[p_{l_1}(\boldsymbol{x}_i), \dots, p_{l_{|L|+1}}(\boldsymbol{x}_i)]^T\}_{i=1}^n. \tag{10}$$

The alignment between $F(\{\boldsymbol{x}\}_n)$ and $\{\boldsymbol{y}\}_m$ is parameterized by $\boldsymbol{\alpha}[\{\boldsymbol{x}\}_n, W] \in \Delta^n$, defined as

$$\boldsymbol{\alpha}[\{\boldsymbol{x}\}_n, W] = \left[ \frac{e^{W(\boldsymbol{x}_1)}}{\sum_{i=1}^n e^{W(\boldsymbol{x}_i)}}, \ldots, \frac{e^{W(\boldsymbol{x}_n)}}{\sum_{i=1}^n e^{W(\boldsymbol{x}_i)}} \right]^T, \tag{11}$$

where $W$ is a network that outputs a scalar for each frame $\boldsymbol{x}_i$. Using the framework built in Section 4.2.1 (with $r = 1$ and $C = C_e$, where $C_e$ is the cross-entropy) to predict $\{\boldsymbol{y}\}_m$ from $\{\boldsymbol{x}\}_n$, we train both $W$ and $F$ by minimizing the OTTC objective

$$\mathcal{L}_{OTTC} = - \sum_{i,j=1}^{n,m} \boldsymbol{\gamma}_n^{m,\boldsymbol{\beta}_m}(\boldsymbol{\alpha}[\{\boldsymbol{x}\}_n, W])_{i,j} \cdot \log p_{\boldsymbol{y}_j}(\boldsymbol{x}_i). \tag{12}$$

The choice of the cross-entropy $C_e$ as the cost function arises naturally from the probabilistic encoding of the predicted output of $F$ and the one-hot encoding of the target sequence. Additionally, since $C_e$ is differentiable, it makes the OTTC loss differentiable with respect to $F$, while the differentiability of the OTTC with respect to $W$ stems from the differentiability of $\boldsymbol{\gamma}_n^{m,\boldsymbol{\beta}_m}$ with respect to its input $\boldsymbol{\alpha}[\{\boldsymbol{x}\}_n, W]$. Thus, by following the gradient of this loss, we jointly learn both the alignment (via $W$) and the classification (via $F$).

**Note:** The notation $\boldsymbol{\gamma}_n^{m,\boldsymbol{\beta}}$ in Eq. 12 is valid in the context of ASR since $n \geq m$.

### 4.4 LINK WITH CTC LOSS

In this section, we contrast the CTC with the proposed OTTC loss. In the context of CTC, we denote by $\mathcal{B}$ the mapping which reduces any sequences by deleting repeated vocabulary (similarly to the previously defined $\mathcal{A}$ mapping in Proposition 5) *and then* deleting the blank token $\phi$ (e.g., $\mathcal{B}(\{GGOO\phi ODD\}) = \{GOOD\}$). The objective of CTC is to maximise the probability of all possible paths $\{\boldsymbol{\pi}\}_n$ of length $n$ through minimizing

$$- \sum_{\{\boldsymbol{\pi}\}_n \in \mathcal{B}^{-1}(\{\boldsymbol{y}\}_m)} \log p(\{\boldsymbol{\pi}\}_n) = - \sum_{\{\boldsymbol{\pi}\}_n \in \mathcal{B}^{-1}(\{\boldsymbol{y}\}_m)} \log \prod_{i=1}^n p(\boldsymbol{\pi}_i), \tag{13}$$

where $\{\boldsymbol{\pi}\} \in L^n$ is an $n$-length sequence and $\mathcal{B}^{-1}(\{\boldsymbol{y}\}_m)$ is the set of all sequences collapsed by $\mathcal{B}$ into $\{\boldsymbol{y}\}_m$.

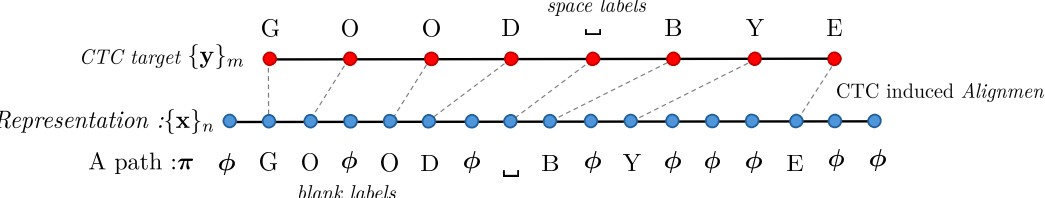

Figure 3: *A CTC alignment.* Here, we illustrate one of the valid alignments for CTC. The CTC loss maximizes the marginal probability over all such possible alignments.

Let us consider a path $\{\boldsymbol{\pi}\}_n \in \mathcal{B}^{-1}(\{\boldsymbol{y}\}_m)$. Such a path can be seen as an alignment (see Figure 3), where $\{\boldsymbol{x}_i\}$ and $\{\boldsymbol{y}_j\}$ are aligned iff $\boldsymbol{\pi}_i = \boldsymbol{y}_j$. By denoting $\boldsymbol{A}_\pi$ as the corresponding discrete monotonic alignment, one can write ($C_e$ represents the Cross-Entropy) :

$$\log p(\{\boldsymbol{\pi}\}_n) = \sum_{i=1}^n \log p_{\boldsymbol{\pi}_i}(\boldsymbol{x}_i) = - \sum_{\substack{i,j=1 \\ (i,j) \in \boldsymbol{A}_\pi}}^{n,m} C_e(\boldsymbol{\pi}_j, \boldsymbol{y}_i) \stackrel{\exists \boldsymbol{\alpha} \in \Delta^n}{=} - \sum_{\substack{i,j=1 \\ \boldsymbol{\gamma}_p^{n,\boldsymbol{\beta}_m}(\boldsymbol{\alpha})_{i,j} > 0}}^{n,m} C_e(\boldsymbol{\pi}_j, \boldsymbol{y}_i). \tag{14}$$

*The last equality arises from Proposition 1 and that $\boldsymbol{A}_\pi$ represents a discrete monotonic alignment.*

The continuous relaxation (i.e. making the problem continuous with respect to the alignment) of the last term in this sequence of equalities results in $-\mathcal{L}_{OTTC}$. Therefore, OTTC can be seen as a relaxation of the probability associated with a single path, enabling a differentiable path search mechanism. Essentially, OTTC optimization focuses on maximizing the probability of exactly one path, in contrast to CTC, which maximizes the probability across all valid paths. Additionally, OTTC does not incentivize paths containing many blank tokens, unlike CTC, as blanks are solely used to separate repeated labels (e.g., consecutive tokens). Instead of relying on a blank token to indicate that a frame $i$ should not be classified, the model can simply set the corresponding weight $\alpha_i$ to 0 (see Figure 2).

## 5 EXPERIMENTAL SETUP

To demonstrate the viability of the proposed OTTC loss framework, we conduct proof-of-concept experiments on the ASR task, which is an important problem from the perspective of seq2seq learning. To this end, we compare results obtained through the OTTC loss framework in terms of the Word Error Rate (WER) with those obtained from a CTC-based model. Note that an efficient batched implementation of OTTC along with the full code to reproduce our experimental results will be made publicly available.

**Datasets.** We conduct our experiments on popular open-source datasets, i.e., the LibriSpeech (Panayotov et al., 2015) and AMI (Carletta et al., 2005) datasets. LibriSpeech is an English read-speech corpus derived from audiobooks, containing 1000 hours of data. For our experiments on this dataset, we train models on the official 100-hour, 360-hour, and 960-hour splits, and report results on the two official test sets. AMI is an English spontaneous meeting speech corpus, which differs significantly from read-speech. For our experiments on this dataset, we train models on the individual head microphone (IHM) split comprising 80 hours of audio, and report results on the official dev and eval sets.

**Model architecture.** We use the 300M parameter version of the well-known XLS-R model (Babu et al., 2021) as the base model for acoustic embeddings in all the experiments conducted in this work. The XLS-R is a self-supervised model pre-trained on 436K hours of unlabeled multilingual speech from 128 different languages. For the baseline CTC-based models, we stack a dropout layer followed by a linear layer for logits prediction, termed the *logits prediction head*. For the proposed OTTC loss based models, we use a dropout and a linear layer (identical to the baseline) for logits prediction. In addition, as described in Section 4.3, we apply a dropout layer followed by two linear layers on top of the XLS-R model for OT weight prediction, with a GeLU (Hendrycks & Gimpel, 2016) non-linearity in between, termed the *OT weights prediction head*. Note that the output from the XLS-R model is used as input for both the logit and OT weight prediction heads, and the entire model is trained using the OTTC loss.

**Training details.** In all our experiments, we use the AdamW optimizer (Loshchilov & Hutter, 2019) for training. For LibriSpeech, the initial learning rate is set to $lr = 2e^{-4}$, with a linear warm-up for the first 500 steps followed by a linear decay until the end of training. For AMI, the initial learning rate is set to $lr = 1.25e^{-3}$, with a linear warm-up during the first $10\%$ of the steps, also followed by linear decay. We train both CTC-based and OTTC-based models for 40 epochs, reporting the test set WER at the final epoch. In our OTTC-based models, both the logits and OT weight prediction heads are trained for the first 30 epochs. During the final 10 epochs, the OT weight prediction head is fixed, while training continues on the logits prediction head. For experiments on the LibriSpeech dataset, we use character-level tokens to encode text. Given the popularity of subword-based units for encoding text (Sennrich et al., 2016), we sought to observe the behavior of OTTC-based models when tokens are subword-based, where a token can contain more than one character. For the experiments on the AMI dataset, we use the SentencePiece tokenizer (Kudo & Richardson, 2018) to train subwords from the training text. Greedy decoding is used for both the CTC and OTTC models to generate the hypothesis text.

**Choice of label weights ($\boldsymbol{\beta}_q$).** To simplify the training setup for our OTTC-based models, we use a fixed and uniform $\boldsymbol{\beta}_q$ (see Sections 4.2 & 4.3), where the length $q$ of $\boldsymbol{\beta}$ is equal to the total number of tokens in the text after augmenting with the blank ($\phi$) label between repeating characters.

Table 1: WER(%) comparison between the CTC loss-based ASR model and our proposed OTTC loss-based ASR model. On the LibriSpeech dataset, models are trained using the three official training splits with varying amounts of supervised data, and results are reported on the two official test sets. For the AMI dataset, models are trained on the IHM split, and results are reported on both the dev and eval sets. Note that for WER, lower is better.

| Model | 100h-LibriSpeech | | 360h-LibriSpeech | | 960h-LibriSpeech | | AMI-IHM | |
|---|---|---|---|---|---|---|---|---|
| | test-clean | test-other | test-clean | test-other | test-clean | test-other | dev | eval |
| CTC | 4.93 | 12.09 | 3.53 | 10.04 | 2.9 | 7.46 | 15.8 | 13.9 |
| OTTC | 7.43 | 17.34 | 5.19 | 13.49 | 4.24 | 10.36 | 18.5 | 16.8 |

## 6 RESULT AND DISCUSSION

We start by analyzing the performance of the considered models on the LibriSpeech dataset, with the results reported in Table 1. Using the 100-hour split for training, the OTTC model achieves a WER of $7.43\%$ on test-clean, demonstrating remarkable alignment learning capability, even when the OT weights for the labels ($\beta_q$) are uniform and independent of the acoustic embedding information. As we scale the training dataset (100h $\rightarrow$ 360h $\rightarrow$ 960h), we see a monotonic improvement in WER for the proposed OTTC-based models, similarly to the CTC-based models. Although the WERs achieved by the OTTC-based models are higher than the WERs achieved by the CTC-based models, the presented results underscore the experimental validity of the SOTD as a metric and demonstrate that learning a single alignment can yield promising results in E2E ASR.

Next, we conduct experiments on the AMI dataset, which contains spontaneous meeting speech, to understand how effectively the OTTC loss can learn alignment with varying speaking rates while using a fixed and uniform $\beta_q$. From the results shown in Table 1 (last column), the OTTC model achieves encouraging performance on the AMI dataset (albeit not yet as competitive as the performance of the CTC model) highlighting the robustness of our proposed alignment framework. The model effectively adapts to the variability in speaking rates, demonstrating that it can learn accurate alignment even with a $\beta$ independent of acoustic frames.

**Additional insights.** *Training OTTC models.* As described in Section 5, the *OT weights prediction head* ($\alpha$ predictor) remains frozen during the last 10 epochs of training (out of a total of 40 epochs) for the OTTC models. In the 960h-LibriSpeech training setup, we observed a WER of $4.77\%$ at epoch 30 for the OTTC model, resulting in an $11\%$ relative reduction by epoch 40. Interestingly, when the model is trained for the full 40 epochs without freezing the *OT weights prediction head*, no meaningful improvement in WER is observed between epochs 30 and 40. This suggests that the alignment stabilizes early in the training, with the OTTC model learning sufficiently robust alignments by epoch 30. Consequently, further joint optimization of both the alignment and logit prediction may be unnecessary in the later stages, as the alignment undergoes minimal changes beyond that point. However, given the mutual reinforcement between the correctness of alignments and classification in the OTTC loss, we hypothesize that an improved curriculum learning framework (Hacohen & Weinshall, 2019) could further improve ASR performance, which we leave for future work.

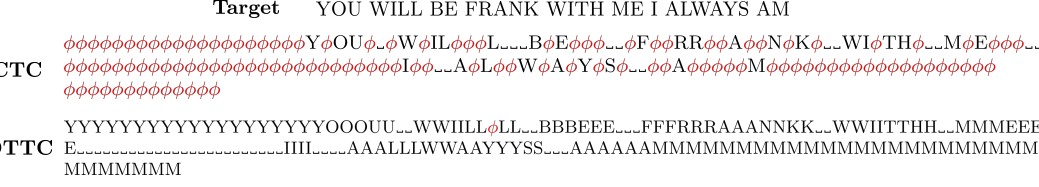

Figure 4: *Comparison of CTC and OTTC alignments.* For CTC, the path with highest probability is shown. CTC shows a high occurrence of blank tokens with sparse non-blank assignments, resulting in peaky behavior. OTTC rarely aligns frames to blank tokens, avoiding this peaky pattern.

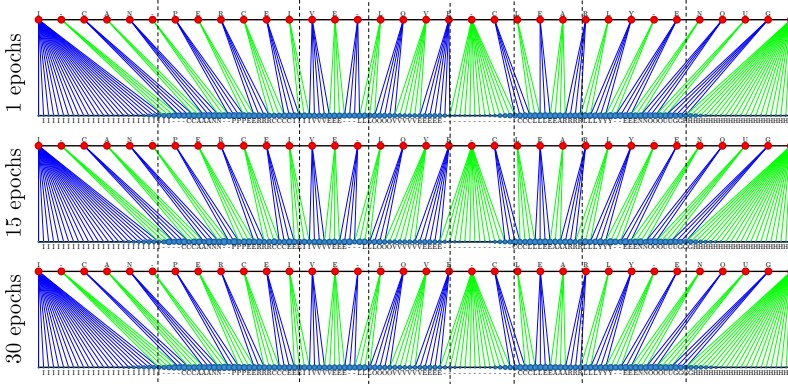

Figure 5: ***Evolution of alignment in the OTTC model during the course of training.*** The red bullets represent elements of the target sequence $\{y\}_m$, while the blue bullets indicate the predicted OT weights for each frame. The size of the blue bullets is proportional to the predicted OT weight.

*Obtained alignments in CTC and OTTC models.* To additionally support our motivations for proposing OTTC, we show an alignment from the CTC- and OTTC-based models in Figure 4. For CTC, it can be seen that the best path aligns most of frames to the blank token, resulting in a peaky behavior Zeyer et al. (2021). In contrast, the OTTC model learns to align nearly all frames, except for one frame involving a repeating character, to non-blank tokens. This effectively mitigates the peaky behavior observed in the CTC model. Note that OTTC allows dropping frames during alignment (see Section 4.4), however, in practice, we observed that only a few frames are dropped. For additional insights, we plot the evolution of the alignment for the OTTC model during the course of training in Figure 5. It is evident that the alignment learned early in the training process remains relatively stable as training progresses. The most notable changes occur at the extremities of the predicted label clusters. This observation led us to the decision to freeze the OT weight predictions for the final 10 epochs, otherwise, even subtle changes in alignment could adversely impact the logits predictions because same base model is shared for predicting both the logits and the alignment OT weights.

In summary, the presented results show that while the proposed OTTC models yield an advantageous performance, there remains a performance gap to CTC models. While we considered fixed label weights ($\{\beta\}_N$) in our experiments, the framework allows for learnable label weights. However, without proper constraints on the minimum values of the label weights, this could lead to a degenerate solution where all acoustic frames align with a random label, causing alignment collapse. We envision that learning label weights with suitable constraints can bridge the performance gap with CTC models. Furthermore, our framework effectively addresses the peaky behavior commonly seen in CTC models, resulting in improved alignments.

## 7 CONCLUSION AND FUTURE WORK

Learning effective sequence-to-sequence mapping along with its corresponding alignment has diverse applications across various fields. Building upon our core idea of modeling the alignment between two sequences as a learnable mapping while simultaneously predicting the target sequence, we define a pseudo-metric known as the Sequence Optimal Transport Distance (SOTD) over sequences. Our formulation of SOTD enables the joint optimization of target sequence prediction and alignment, which is achieved through one-dimensional optimal transport. We theoretically show that the SOTD indeed defines a distance with guaranteed existence of a solution, though uniqueness is not assured. We then derive the Optimal Temporal Transport Classification (OTTC) loss for automatic speech recognition (ASR) where the task is to map acoustic frames to text. Experiments on the LibriSpeech and AMI datasets show that our method achieves encouraging performance in ASR. Importantly, multiple alignment plots for the OTTC model demonstrate that it does not lead to the peaky behavior observed in CTC-based models.

While we use fixed label weights in our experiments, the framework supports learnable label weights, a promising direction for future work. Additionally, exploring alternative curriculum learning strategies between alignment and logits during training could enhance performance. Finally, other sequence-to-sequence tasks could be investigated using the proposed framework, particularly those involving the alignment of multiple sequences, such as audio, video, and text.

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

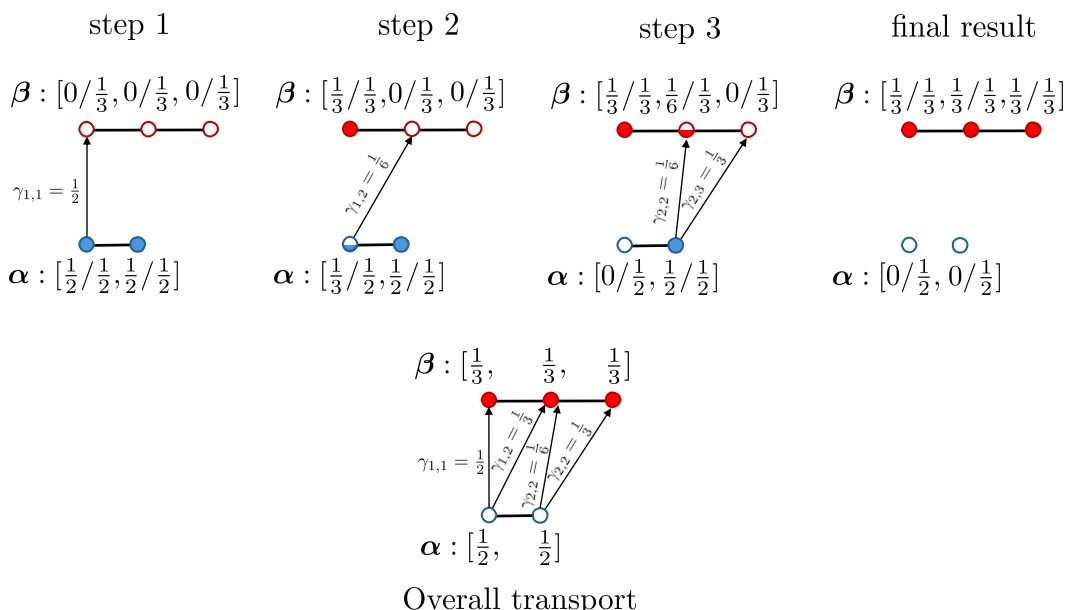

Figure 6: **1D OT transport computation.** Illustration of the optimal transport process, computed iteratively by transferring probability mass from the smallest bins to the largest.

## A APPENDIX

### A.1 ALGORITHM AND IMPLEMENTATION DETAILS

#### A.1.1 ALIGNMENT COMPUTATION

The algorithm to compute $\gamma_n^{m,\beta}$ is given in Algorithm 1. This algorithm computes the 1D optimal transport between $\mu[\alpha, n]$ and $\nu[\beta, m]$, exploiting the monotonicity of transport in this dimension. To do so the first step consist in sorting the bins which has the complexity $O(n \log n) + O(m \log m) = O(\max(n,m) \log \max(n,m))$. Then we transfer the probability mass from one distribution to another, moving from the smallest bins to the largest. A useful way to visualize this process is by imagining that the bins of $\mu$ each contain a pot with a volume of $a_i$ filled with water, while the bins of $\nu$ each contain an empty pot with a volume of $b_j$. The goal is to fill the empty pots of $\nu$ using the water from the pots of $\mu$. At any given step of the process, we always transfer water from the smallest non-empty pot of $\mu$ to the smallest non-full pot of $\nu$. The volume of water transferred from $i$ to $j$ is denoted by $\gamma_{i,j}$. An example of this process is provided in Figure 6.

In the worst case, this process requires $O(n+m)$ comparisons. However, since the bins are already sorted in SOTD, the overall complexity remains $O(n+m) = O(\max(n,m))$. In practice, this algorithm is not directly used in this work, as we never compute optimal transport solely; it is provided here to illustrate that the dependencies of $\gamma_n^{m,\beta}$ on $\alpha$ are explicit, making it differentiable with respect to $\alpha$. An efficient batched implementation version for computing SOTD will be released soon.

### A.2 PROPERTIES OF OTTC

Here can be found proof and more insight about the properties of SOTD, $\mathcal{S}_r$.

#### A.2.1 LEMMA 1 : BIJECTIVITY

**Proof of Lemma 1.** *Surjectivity*: The surjectivity come from definition of $\Gamma^{*,\beta}[n]$. *Injectivity*: Suppose $\gamma_n^{m,\beta}(\alpha) = \gamma_n^{m,\beta}(\sigma)$, so $\alpha = [\sum_{j=1}^m \gamma_n^{m,\beta}(\alpha)_{i,j}, \ldots, \sum_{j=1}^m \gamma_n^{m,\beta}(\alpha)_{i,j}]^T =$

---

**Algorithm 1** : Transport Computation - $\boldsymbol{\gamma}_n^{m,\boldsymbol{\beta}}(\boldsymbol{\alpha})$ -

---

**Ensure:** Compute $\boldsymbol{\gamma}_n^{m,\boldsymbol{\beta}}(\boldsymbol{\alpha})$.
**Require:** $\boldsymbol{\alpha} \in \mathbb{R}^n$.
  Set $\boldsymbol{\gamma} \in \mathbb{R}^{n \times m} = \mathbf{0}_{n \times m}$.
  Set $i, j = 0$.
  **while** $T == True$ **do**
    **if** $\alpha_i < \beta_j$ **then**
      $\boldsymbol{\gamma}_{i,j} = \beta_j - \alpha_i$
      $i = i + 1$
      **if** $i == n$ **then**
        $T = false$
      $\beta_j = \beta_j - \alpha_i$
    **else**
      $\boldsymbol{\gamma}_{i,j} = \alpha_i - \beta_j$
      $j = j + 1$
      **if** $j == m$ **then**
        $T = false$
      $\alpha_i = \alpha_i - \beta_j$
  **return** $\boldsymbol{\gamma}$

---

$[\sum_{j=1}^m \boldsymbol{\gamma}_n^{m,\boldsymbol{\beta}}(\boldsymbol{\sigma})_{i,j}, \ldots, \sum_{j=1}^m \boldsymbol{\gamma}_n^{m,\boldsymbol{\beta}}(\boldsymbol{\sigma})_{i,j}]^T = \boldsymbol{\sigma}$ (because $\boldsymbol{\gamma}_n^{m,\boldsymbol{\beta}}(\boldsymbol{\alpha}) \in \Gamma^{\boldsymbol{\alpha},\boldsymbol{\beta}}$ and $\boldsymbol{\gamma}_n^{m,\boldsymbol{\beta}}(\boldsymbol{\sigma}) \in \Gamma^{\boldsymbol{\sigma},\boldsymbol{\beta}}$), which conclude the proof.

### A.2.2 PROPOSITION 1 : DISCRETE MONOTONIC ALIGNMENT APPROXIMATION EQUIVALENCE.

**Proof of proposition 1**. Let's consider the following proposition $P(k)$ :

$$P(k) : \exists \boldsymbol{\alpha}^i \in \Delta^n, \forall i, \forall j \leq k, (i,j) \in \boldsymbol{A} \iff \boldsymbol{\gamma}_n^{m,\boldsymbol{\beta}}(\boldsymbol{\alpha}^i)_{i,j} > 0. \tag{15}$$

**Initialisation -** $P(1)$. $P(1)$ is true. Consider the set $E_1 = \{j \in [\![1,m]\!] \mid (1,j) \in \mathbf{A}\}$, which can be written as $E_1 = \{1, 2, \ldots, \max(E_1)\}$ since $A$ is a discrete monotonic alignment. Define $\boldsymbol{\alpha}^1 = [\sum_{j \in E_1} \beta_j, \ldots]^T$, where the remaining coefficients are chosen to sum to 1.

Since the alignment $\boldsymbol{\gamma}_n^{m,\boldsymbol{\beta}}$ is computed monotonically (see Appendix A.1.1), $\boldsymbol{\gamma}_n^{m,\boldsymbol{\beta}}(\boldsymbol{\alpha}^1)_{1,j} > 0$ if and only if $\alpha_1^1 \leq \beta_1 + \cdots + \beta_j$, which corresponds exactly to the set of indices $j \in E_1$, i.e., the aligned indices in $\mathbf{A}$. This proves $P(1)$.

**Heredity -** $P(k) \Rightarrow P(k+1)$. The proof follows similarly to $P(1)$. However two cases need to be considered :

- When $(k+1, \max(E_k)) \in \mathbf{A}$, in this cases we must consider $E_{k+1} = \{j \in [\![1,m]\!] \mid (k+1, j) \in \mathbf{A}\} = \{\max(E_k) = \min(E_{k+1}), \min(E_{k+1}) + 1, \ldots, \max(E_{k+1})\}$ (because $\boldsymbol{\beta}$ has no components) and define $\boldsymbol{\alpha}^{k+1} = [\alpha_1^1, \ldots, \alpha_k^k - \frac{\beta_{\max(E_k)}}{2}, \sum_{j \in E_{k+1}} \beta_j - \frac{\beta_{\max(E_k)}}{2}, \ldots]^T$, where the remaining parameters are chosen to sum to 1.

- When $(k+1, \max(E_k)) \notin \mathbf{A}$, we must consider $E_{k+1} = \{j \in [\![1,m]\!] \mid (k+1, j) \in \mathbf{A}\} = \{\max(E_k) \neq \min(E_{k+1}), \min(E_{k+1}) + 1, \ldots, \max(E_{k+1})\}$ (because $\boldsymbol{\beta}$ has no components) and define $\boldsymbol{\alpha}^{k+1} = [\alpha_1^1, \ldots, \alpha_k^k, \sum_{j \in E_{k+1}} \beta_j, \ldots]^T$, where the remaining parameters are chosen to sum to 1.

By induction, the proposition holds for all $n$. Therefore, Proposition 1 (i.e., $P(n)$) is true. An $\boldsymbol{\alpha}$ verifying the condition is :

$$\boldsymbol{\alpha} = [\alpha_1^1, \ldots, \alpha_n^n]^T$$

### A.2.3 PROPOSITION 2 : VALIDITY OF SOTD DEFINITION

**Proof of proposition 2.** Since $\boldsymbol{\gamma}_n^{m,\boldsymbol{\beta}}$ is differentiable so continuous, it follows that $\boldsymbol{\alpha} \mapsto \sum_{i,j=1}^{n,m} \boldsymbol{\gamma}_n^{m,\boldsymbol{\beta}}(\boldsymbol{\alpha})_{i,j} \cdot C(\boldsymbol{x}_i, \boldsymbol{y}_j)$ is continuous over $\Delta^n$. Given that $\Delta^n$ is a compact set and every continuous function on a compact space is bounded and attains its bounds, the existence of an optimal solution $\boldsymbol{\alpha}^*$ follows.

**Non-unicity of the solution.** The non unicity come from that if their is a solution $\boldsymbol{\alpha}^*$ and two integer $k$, $l$ such that $\boldsymbol{\gamma}_n^{m,\boldsymbol{\beta}}(\boldsymbol{\alpha}^*)_{k,l} \geq \epsilon > 0$ and $\boldsymbol{\gamma}_n^{m,\boldsymbol{\beta}}(\boldsymbol{\alpha}^*)_{k+1,l} \geq \epsilon > 0$ and $C(\boldsymbol{x}_k, \boldsymbol{y}_l) = C(\boldsymbol{x}_{k+1}, \boldsymbol{y}_l)$, therefore the transport $\hat{\gamma}$ such that :

- $\forall i \in [\![1, n]\!], j, \in [\![1, m]\!], (i,j) \neq (k,l), \hat{\gamma}_{i,j} = \boldsymbol{\gamma}_n^{m,\boldsymbol{\beta}}(\boldsymbol{\alpha}^*)_{i,j}$.

- $\hat{\gamma}_{k,l} = \boldsymbol{\gamma}_n^{m,\boldsymbol{\beta}}(\boldsymbol{\alpha}^*)_{k,l} - \epsilon/2$

- $\hat{\gamma}_{k+1,l} = \boldsymbol{\gamma}_n^{m,\boldsymbol{\beta}}(\boldsymbol{\alpha}^*)_{k+1,l} + \epsilon/2$

Let's denote $\boldsymbol{\sigma} = \{\boldsymbol{\gamma}_n^{m,\boldsymbol{\beta}}\}^{-1}(\hat{\gamma}_{i,j})$. First $\boldsymbol{\sigma} \neq \boldsymbol{\alpha}$ because $\sigma_k = \sum_{l=1}^m \hat{\gamma}_{k,l} = \sum_{l=1}^m \boldsymbol{\gamma}_n^{m,\boldsymbol{\beta}}(\boldsymbol{\alpha}^*)_{k,l} - \epsilon/2 = \alpha_k^* - \epsilon/2$. Second, it's clear that $\sum_{i,j=1}^{n,m} \boldsymbol{\gamma}_n^{m,\boldsymbol{\beta}}(\boldsymbol{\alpha}^*)_{i,j} \cdot C(\boldsymbol{x}_i, \boldsymbol{y}_j) = \sum_{i,j=1}^{n,m} \boldsymbol{\gamma}^{m,\boldsymbol{\beta}_n}(\boldsymbol{\sigma})_{i,j} \cdot C(\boldsymbol{x}_i, \boldsymbol{y}_j)$. Then $\boldsymbol{\sigma}$ is distinct solution.

### A.2.4 PROPOSITION 3 : SOTD IS A PSEUDO METRIC

**Proof of proposition 3.** *Pseudo-separation.* It's clear that $\mathcal{S}_r(\{\boldsymbol{x}\}_n, \{\boldsymbol{x}\}_n) = 0$, this value is attained for $\alpha^* = \boldsymbol{\beta}_n$; where the corresponding alignment $\boldsymbol{\gamma}_n^{n,\boldsymbol{\beta}_n}(\boldsymbol{\alpha}^*)$ corresponds to a one-to-one alignment. Since the two sequences are identical, all the costs are zero.

*Symmetry*. We have $\mathcal{S}_r(\{\boldsymbol{x}\}_n, \{\boldsymbol{y}\}_m m) = \mathcal{S}_r(\{\boldsymbol{y}\}_m, \{\boldsymbol{x}\}_n)$ because the expression for $\mathcal{S}_r$ in Eq. 7 is symmetric. Specifically, because $C$ is symmetric as it is a metric.

*Triangular inequality.* Consider three sequences $\{\boldsymbol{x}\}_n, \{\boldsymbol{y}\}_m$ and $\{\boldsymbol{z}\}_o$. Let $p = \max(n,m)$, $q = \min(n,m)$, $u = \max(m,o)$, $v = \min(m,o)$. Define the optimal alignments $\boldsymbol{\gamma}_p^{q,\boldsymbol{\beta}_q}(\boldsymbol{\alpha}^*)$ between $\{\boldsymbol{x}\}_n$ and $\{\boldsymbol{y}\}_m$; and $\boldsymbol{\gamma}_u^{v,\boldsymbol{\beta}_v}(\boldsymbol{\rho}^*)$ between $\{\boldsymbol{y}\}_m$ and $\{\boldsymbol{z}\}_o$. $\forall i \in [\![1,n]\!], \forall j, k \in [\![1,m]\!], \forall l \in [\![1,o]\!]$, we define :

$$\gamma_{i,j}^{xy} = \begin{cases} \boldsymbol{\gamma}_p^{q,\boldsymbol{\beta}_q}(\boldsymbol{\alpha}^*)_{i,j} & \text{if } n \geq m \\ \boldsymbol{\gamma}_p^{q,\boldsymbol{\beta}_q}(\boldsymbol{\alpha}^*)_{j,i} & \text{otherwise.} \end{cases} \tag{16}$$

$$\gamma_{k,l}^{yz} = \begin{cases} \boldsymbol{\gamma}_u^{v,\boldsymbol{\beta}_v}(\boldsymbol{\rho}^*)_{k,l} & \text{if } k \geq l \\ \boldsymbol{\gamma}_u^{v,\boldsymbol{\beta}_v}(\boldsymbol{\rho}^*)_{l,k} & \text{otherwise.} \end{cases} \tag{17}$$

$$\gamma_{j,k}^{yy} = \boldsymbol{\gamma}_p^{q,\boldsymbol{\sigma}^*}(\boldsymbol{\beta}_q)_{j,k} \tag{18}$$

and we define :

$$b_j = \begin{cases} \sum_{i=1}^n \gamma_{i,j}^{xy} & \text{if } > 0 \\ 1 & \text{otherwise.} \end{cases} \tag{19}$$

$$c_k = \begin{cases} \sum_{l=1}^o \gamma_{k,l}^{yz} & \text{if } > 0 \\ 1 & \text{otherwise.} \end{cases} \tag{20}$$

So $\gamma^{xy}$ is the optimal transport between $\mu[\boldsymbol{\alpha}^*, p]$ and $\nu[\boldsymbol{\beta}_q, q]$; $\gamma^{yy}$ is the optimal transport between $\mu[\boldsymbol{\beta}_q, q]$ and $\nu[\boldsymbol{\sigma}^*, u]$ and $\gamma^{yz}$ is the optimal transport between $\mu[\boldsymbol{\sigma}^*, u]$ and $\nu[\boldsymbol{\beta}_v, v]$, since in 1D optimal transport can be composed, the composition $\frac{\gamma_{i,j}^{xy} \gamma_{j,k}^{yy} \gamma_{k,l}^{yz}}{b_j c_k}$ is an optimal transport between $\mu[\boldsymbol{\alpha}^*, p]$ and $\nu[\boldsymbol{\beta}_v, v]$. Therefore by bijectivity of $\boldsymbol{\gamma}_{\max(p,v)}^{\min(p,v),\boldsymbol{\beta}_{\min(p,v)}}$, there is a $\boldsymbol{\theta} \in \mathbb{R}^{\max(p,v)}$ such that :

$$\gamma_{\max(p,v)}^{\min(p,v),\boldsymbol{\beta}_{\min(p,v)}}(\boldsymbol{\theta}) = \frac{\gamma_{i,j}^{xy}\gamma_{j,k}^{yy}\gamma_{k,l}^{yz}}{b_j c_k} \tag{21}$$

Thus, by the definition of $\mathcal{S}_r(\{\boldsymbol{x}\}_n, \{\boldsymbol{z}\}_o)$:

$$\mathcal{S}_r(\{\boldsymbol{x}\}_n, \{\boldsymbol{z}\}_o) \leq \Big( \sum_{i,l=1}^{n,o} \sum_{j,k=1}^{m,m} \gamma_{\max(p,v)}^{\min(p,v),\boldsymbol{\beta}_{\min(p,v)}}(\boldsymbol{\theta}) \cdot C(\boldsymbol{x}_i, \boldsymbol{z}_l)^r \Big)^{1/r} \tag{22}$$

$$\mathcal{S}_r(\{\boldsymbol{x}\}_n, \{\boldsymbol{z}\}_o) \leq \Big( \sum_{i,l=1}^{n,o} \sum_{j,k=1}^{m,m} \frac{\gamma_{i,j}^{xy}\gamma_{j,k}^{yy}\gamma_{k,l}^{yz}}{b_j c_k} \cdot C(\boldsymbol{x}_i, \boldsymbol{z}_l)^r \Big)^{1/r} \tag{23}$$

$$\mathcal{S}_r(\{\boldsymbol{x}\}_n, \{\boldsymbol{z}\}_o) \leq \Big( \sum_{i,l=1}^{n,o} \sum_{j,k=1}^{m,m} \frac{\gamma_{i,j}^{xy}\gamma_{j,k}^{yy}\gamma_{k,l}^{yz}}{b_j c_k} \cdot (C(\boldsymbol{x}_i, \boldsymbol{y}_j) + C(\boldsymbol{y}_j, \boldsymbol{y}_k) + C(\boldsymbol{y}_k, \boldsymbol{z}_l))^r \Big)^{1/r} \tag{24}$$

Applying the Minkowski inequality:

$$\mathcal{S}_r(\{\boldsymbol{x}\}_n, \{\boldsymbol{z}\}_o) \leq \Big( \sum_{i,l=1}^{n,o} \sum_{j,k=1}^{m,m} \frac{\gamma_{i,j}^{xy}\gamma_{j,k}^{yy}\gamma_{k,l}^{yz}}{b_j c_k} \cdot (C(\boldsymbol{x}_i, \boldsymbol{y}_j))^r \Big)^{1/r} + \tag{25}$$

$$\Big( \sum_{i,l=1}^{n,o} \sum_{j,k=1}^{m,m} \frac{\gamma_{i,j}^{xy}\gamma_{j,k}^{yy}\gamma_{k,l}^{yz}}{b_j c_k} \cdot (C(\boldsymbol{y}_j, \boldsymbol{y}_k))^r \Big)^{1/r} + \tag{26}$$

$$\Big( \sum_{i,l=1}^{n,o} \sum_{j,k=1}^{m,m} \frac{\gamma_{i,j}^{xy}\gamma_{j,k}^{yy}\gamma_{k,l}^{yz}}{b_j c_k} \cdot (C(\boldsymbol{y}_k, \boldsymbol{z}_l))^r \Big)^{1/r} \tag{27}$$

Then :

$$\mathcal{S}_r(\{\boldsymbol{x}\}_n, \{\boldsymbol{z}\}_o) \leq \Big( \sum_{i,j=1}^{n,m} \gamma_{i,j}^{xy} \cdot C(\boldsymbol{x}_i, \boldsymbol{y}_j)^r \Big)^{1/r} + \tag{28}$$

$$\Big( \sum_{j,k=1}^{m,m} \gamma_{j,k}^{yy} \cdot C(\boldsymbol{y}_j, \boldsymbol{y}_k)^r \Big)^{1/r} + \tag{29}$$

$$\Big( \sum_{k,l=1}^{m,o} \gamma_{k,l}^{yz} \cdot C(\boldsymbol{y}_k, \boldsymbol{z}_l)^r \Big)^{1/r} \tag{30}$$

By definition :

$$\mathcal{S}_r(\{\boldsymbol{x}\}_n, \{\boldsymbol{z}\}_o) \leq \mathcal{S}_r(\{\boldsymbol{x}\}_n, \{\boldsymbol{y}\}_m) + \mathcal{S}_r(\{\boldsymbol{y}\}_m, \{\boldsymbol{y}\}_m) + \mathcal{S}_r(\{\boldsymbol{y}\}_m, \{\boldsymbol{z}\}_o) \tag{31}$$

So finally since $\mathcal{S}_r(\{\boldsymbol{y}\}_m, \{\boldsymbol{y}\}_m) = 0$, the triangular inequality holds :

$$\mathcal{S}_r(\{\boldsymbol{x}\}_n, \{\boldsymbol{z}\}_o) \leq \mathcal{S}_r(\{\boldsymbol{x}\}_n, \{\boldsymbol{y}\}_m) + \mathcal{S}_r(\{\boldsymbol{y}\}_m, \{\boldsymbol{z}\}_o). \tag{32}$$

This concludes the proof.

### A.2.5 PROPOSITION 4 : NON-SEPARATION CONDITION

*Proof.* Suppose $\mathcal{S}_r(\{\boldsymbol{x}\}_n, \{\boldsymbol{y}\}_m) = 0$, and $\mathcal{A}(\mathcal{P}_{\alpha^*}(\{\boldsymbol{x}\}_n)) \neq \mathcal{A}(\{\boldsymbol{y}\}_n)$. So :

$$\sum_{i,j=1}^{n,m} \boldsymbol{\gamma}_n^{m,\boldsymbol{\beta}}(\boldsymbol{\alpha}^*)_{i,j} \cdot C(\boldsymbol{x}_i, \boldsymbol{y}_j)^r = 0 \tag{33}$$

Let $\mathcal{A}_{\{\boldsymbol{x}\}_n}$ denote the aggregation operator on $\Delta^n$, which groups indices where consecutive elements in $\{\boldsymbol{x}\}_n$ are identical (i.e, $\mathcal{A}([\ldots, \alpha_i, \ldots, \alpha_{i+k}, \ldots]^T) = [\ldots, \alpha_i + \cdots + \alpha_{i+k}, \ldots]^T$ iff $\boldsymbol{x}_i = \cdots = \boldsymbol{x}_{i+k}$). By expanding the right term, we show that; $\forall \boldsymbol{\alpha} \in \mathbb{R}$. :

$$\sum_{i,j=1}^{n,m} \boldsymbol{\gamma}_n^{m,\boldsymbol{\beta}}(\boldsymbol{\alpha})_{i,j} \cdot C(\boldsymbol{x}_i, \boldsymbol{y}_j)^r = \sum_{i,j=1}^{n,m} \boldsymbol{\gamma}_n^{m,\mathcal{A}_{\{\boldsymbol{y}\}_m}(\boldsymbol{\beta})}(\mathcal{A}_{\{\boldsymbol{x}\}_n}(\boldsymbol{\alpha}))_{i,j} \cdot C(\mathcal{A}(\mathcal{P}_{\boldsymbol{\alpha}}(\{\boldsymbol{x}\}_n)), \mathcal{A}(\{\boldsymbol{y}\}_n))^r \tag{34}$$

Therefore :

$$\sum_{i,j=1}^{n,m} \boldsymbol{\gamma}_n^{m,\mathcal{A}_{\{\boldsymbol{y}\}_m}(\boldsymbol{\beta})}(\mathcal{A}_{\mathcal{P}_{\alpha}\{\boldsymbol{x}\}_n}(\boldsymbol{\alpha}^*))_{i,j} \cdot C(\mathcal{A}(\mathcal{P}_{\alpha^*}(\{\boldsymbol{x}\}_n)), \mathcal{A}(\{\boldsymbol{y}\}_n))^r = 0 \tag{35}$$

Since $\mathcal{A}(\mathcal{P}_{\alpha^*}(\{\boldsymbol{x}\}_n)) \neq \mathcal{A}(\{\boldsymbol{y}\}_n)$ their is a $k \in [\![1, m[\![$ such that :

$$\forall k' < k, \mathcal{A}(\{\boldsymbol{x}\}_n)_{k'} = \mathcal{A}(\{\boldsymbol{y}\}_n)_{k'} \quad \textbf{and} \quad \mathcal{A}(\{\boldsymbol{x}\}_n)_k \neq \mathcal{A}(\{\boldsymbol{y}\}_n)_k \tag{36}$$

Because the optimal alignment is monotonous and lead to a 0 cost, necessarily :

$$\forall k' < k, \mathcal{A}_{\mathcal{P}_{\alpha}(\{\boldsymbol{x}\}_n)}(\boldsymbol{\alpha}^*)_{k'} = \mathcal{A}_{\{\boldsymbol{y}\}_m}(\boldsymbol{\beta})_{k'} \tag{37}$$

which is the only way to have alignment between the $k$ first element which led to 0 cost. Because of the monotonicity of $\boldsymbol{\gamma}_n^{m,\mathcal{A}_{\{\boldsymbol{y}\}_m}(\boldsymbol{\beta})}(\mathcal{A}_{\mathcal{P}_{\alpha}\{\boldsymbol{x}\}_n}(\boldsymbol{\alpha}^*))$ the next alignment $(s, t)$ is between the next element with a non zeros weights for both sequences. Since $\beta$ has non zero component and by the definition of $\mathcal{P}_{\alpha}$, $s = k$ and $t = k$. Therefore the term $\boldsymbol{\gamma}_n^{m,\mathcal{A}_{\{\boldsymbol{y}\}_m}(\boldsymbol{\beta})}(\mathcal{A}_{\mathcal{P}_{\alpha^*}(\{\boldsymbol{x}\}_n)}(\boldsymbol{\alpha}^*))_{k,k}$ is non null and the term :

$$\boldsymbol{\gamma}_n^{m,\mathcal{A}_{\{\boldsymbol{y}\}_m}(\boldsymbol{\beta})}(\mathcal{A}_{\mathcal{P}_{\alpha}\{\boldsymbol{x}\}_n}(\boldsymbol{\alpha}^*))C(\mathcal{A}(\mathcal{P}_{\alpha^*}(\{\boldsymbol{x}\}_n), \mathcal{A}(\{\boldsymbol{y}\}_n)_k)$$

belong to the sum in depicted in Eq. 35. So $C(\mathcal{A}(\mathcal{P}_{\alpha^*}(\{\boldsymbol{x}\}_n)), \mathcal{A}(\{\boldsymbol{y}\}_n)_k) = 0$ i.e., $\mathcal{A}(\mathcal{P}_{\alpha^*}(\{\boldsymbol{x}\}_n)) = \mathcal{A}(\{\boldsymbol{y}\}_n)_k$ because $C$ is separated. Here a contradiction so we can conclude that :

$$\mathcal{A}(\mathcal{P}_{\alpha^*}(\{\boldsymbol{x}\}_n)) = \mathcal{A}(\{\boldsymbol{y}\}_n)$$

.

## A.3 SUPPLEMENTARY EXPERIMENTAL INSIGHTS

### A.3.1 EXPANDED RESULTS

To further evaluate the proposed OTTC framework, we experimented with Wav2Vec2-large (Baevski et al., 2020) as the pre-trained model instead of XLSR, following the same LibriSpeech experimental setup described in Section 5. The results shown in Table 2 indicate that using this pre-trained model further narrows the performance gap between OTTC and CTC.

Table 2: WER(%) comparison between the CTC loss-based ASR model and our proposed OTTC loss-based ASR model using Wav2Vec2-large as the pretrained model for the LibriSpeech dataset. Models are trained using the three official training splits with varying amounts of supervised data. Results are reported for the two official test sets.

| Model | 100h-LibriSpeech | | 360h-LibriSpeech | | 960h-LibriSpeech | |
|-------|-----------|------------|-----------|------------|-----------|------------|
| | test-clean | test-other | test-clean | test-other | test-clean | test-other |
| CTC | 3.36 | 7.36 | 2.77 | 6.58 | 2.20 | 5.23 |
| OTTC | 3.77 | 8.55 | 3.00 | 7.44 | 2.52 | 6.16 |

### A.3.2 ABLATION STUDIES

This section explores the effects of various design choices and configurations on the performance of the proposed OTTC framework and provides additional insights on its comparison to soft-DTW.

**Training with single-path alignment from CTC.** A relevant question that arises is whether the gap between the OTTC and CTC models arises from the use of a single alignment in OTTC rather than marginalizing over all possible alignments. To investigate this, we conducted a comparison with a single-path alignment approach. Specifically, we first obtained the best path (forced alignment using the Viterbi algorithm) from a trained CTC-based model on the same dataset. A new model was then trained to learn this single best path using Cross-Entropy. On the 360-hour LibriSpeech setup with Wav2Vec2-large as the pre-trained model, this single-path approach achieved a WER of 7.04% on the test-clean set and 13.03% on the test-other set. In contrast, under the same setup, the OTTC model achieved considerably better results, with a WER of 3.00% on test-clean and 7.44% on test-other (see Table 2). These findings indicate that the OTTC model is effective with learning a single alignment, which may be sufficient for achieving competitive ASR performance.

**Fixed OT weights prediction ($\alpha$).** We conducted an additional ablation experiment where we replaced the learnable *OT weight prediction head* with fixed and uniform OT weights ($\alpha$). This approach removes the model's ability to search for the best path, assigning instead a frame to the same label during training. Consequently, the model loses the localization of the text-tokens in the audio. For this experiment, we used the 360-hour LibriSpeech setup with Wav2Vec2-large as the pre-trained model. The results show a WER of 3.51% on test-clean, compared to 2.77% for CTC and 3.00% for OTTC with learnable OT weights. On test-other, the WER was 8.24%, compared to 6.58% for CTC and 7.44% for OTTC with learnable OT weights. These results demonstrate that while using fixed OT weights leads to a slight degradation in performance, the localization property is completely lost, highlighting the importance of learnable OT weights for preserving both performance and localization in the OTTC model.

**Impact of freezing OT weights prediction head across epochs.** In our investigations so far, we arbitrarily selected the number of epochs for which the *OT weights prediction head* ($\alpha$ predictor) remained frozen (see Section 6), as a hyperparameter without any tuning. To further understand its impact, we conducted additional experiments on the 360h-LibriSpeech setup using the Wav2Vec2-large model while freezing the *OT weights prediction head* for the last 5 and 15 epochs. When frozen for the last 5 epochs, we achieve a WER of 3.01%, whereas when frozen for the last 15 epochs, the WER is 3.10%. As shown in the Table 2, freezing the OT head for the last 10 epochs results in a WER of 3.00%. Based on these results, it appears that the model's performance doesn't change considerably when the model is trained for a few more epochs after freezing the alignment part of the OTTC model.

**Learnable $\beta$.** To show the importance of making $\beta$ learnable, we first experiment with learning $\beta$ using a trainable transformer decoder layer with tokenized reference text as input. We observe a degenerate solution in which all label weights ($\beta$) are assigned to a single token while all other tokens receive zero label weights, resulting in a WER of 100%. Intuitively, this behavior is to be expected because the model can learn this shortcut, which still minimizes the OTTC loss (the loss goes to zero) as there are no constraints in the loss to prevent it. Next, we impose a constraint on the learnable $\beta$ values, ensuring they cannot fall below a certain threshold. However, we observe a slight

Table 3: Alignment performance metrics for CTC and OTTC models, including Precision, Recall, F1 score, and Intersection Duration Ratio.

| Model | Precision (%) | Recall (%) | F1 Score (%) | Intersection Duration Ratio (%) |
|-------|---------------|------------|--------------|----------------------------------|
| CTC | 84.26 | 83.62 | 83.94 | 17.19 |
| OTTC | 84.77 | 84.85 | 84.81 | 42.12 |

degradation in performance, with around 1% degradation in WER for the 360-hour LibriSpeech setup.

**Oracle experiment.** We believe that the proposed OTTC framework has the potential to outperform CTC models by making $\beta$ learnable with suitable constraints or by optimizing the choice of static $\beta$. To illustrate this potential, we conduct an oracle experiment where we first force-align audio frames and text tokens using a CTC-based model trained on the same data. This alignment is then used to calculate the $\beta$ values. For example, given the target sentence $YES$ and the best valid path from the Viterbi algorithm ($\phi Y \phi \phi EES$), we re-labeled it to ($\phi Y \phi ES$) and set $\beta = [1/7, 1/7, 2/7, 2/7, 1/7]$. This approach enabled OTTC to learn a uniform distribution for $\alpha$, mimicking CTC's highest probability path. As a result, in both the 100h-LibriSpeech and 360h-LibriSpeech setups, the OTTC model converged much faster and matched the performance of CTC. This experiment underscores the critical role of $\beta$, suggesting that a better strategy for its selection or training will lead to further improvements.

**Comments on soft-DTW.** In soft-DTW, only the first and last elements of sequences are guaranteed to align, while all in-between frames or targets may be ignored; i.e., there is no guarantee that soft-DTW will yield a discrete monotonic alignment. A "powerful" transformation $F$ can map $\mathbf{x}$ to $F(\mathbf{x})$ in such a way that soft-DTW ignores the in-between transformed frames ($F(\mathbf{x})$) and targets ($\mathbf{y}$), which we refer to as a collapse (Section 4.2.1). This is why transformations learned through sequence comparison are typically constrained (e.g., to geometric transformations like rotations) (Vayer et al., 2022). Since transformer architectures are powerful, they are susceptible to collapse as demonstrated by the following experiment we conducted using soft-DTW as the loss function. On the 360h-LibriSpeech setup with Wav2Vec2-large model, the best WER achieved using soft-DTW is 39.43%. In comparison, CTC yields 2.77% whereas the proposed OTTC yields 3.00%. A key advantage of our method is that, by construction, such a collapse is not possible.

### A.3.3 ALIGNMENT ANALYSIS

**Peaky behaviour.** The peaky behavior of CTC models is characterized by a significant proportion of audio frames being assigned to either the blank symbol or the space symbol (non-alphabet symbols) (Zeyer et al., 2021). To quantitatively assess the model's peaky behavior, we calculated the average percentage of audio frames assigned to these two special symbols. For the test-clean set, we found that 60.3% of total frames in CTC models were assigned to these special symbols. In contrast, the OTTC model assigned only 22.9% of frames to these symbols. This highlights the effectiveness of the alignment achieved by our proposed framework, which decisively avoids the extreme peaky behavior exhibited by CTC models.

**Quantitative alignment evaluations.** In addition to the peaky behavior, alignment accuracy serves as another crucial evaluation metric. Since ground truth alignments are unavailable, we assess alignment accuracy through forced alignment, a method previously applied to the AMI dataset (Rastorgueva et al., 2023). Following the methodology in (Rastorgueva et al., 2023), we calculated precision, recall, and F1 score. Note that we only considered word-level timestamps, as they are typically less erroneous than individual phoneme, letter, or sub-word level timestamps. As shown in Table 3, the OTTC model shows better alignment performance. However, these metrics do not provide insight into the predicted duration of the words. To address this, we additionally compute the Intersection Duration Ratio. This metric calculates the duration of the overlap between the reference and predicted word segments, dividing it by the total reference duration of those words. The results are shown in Table 3. This results highlight that, on average, the CTC model either predicts the start

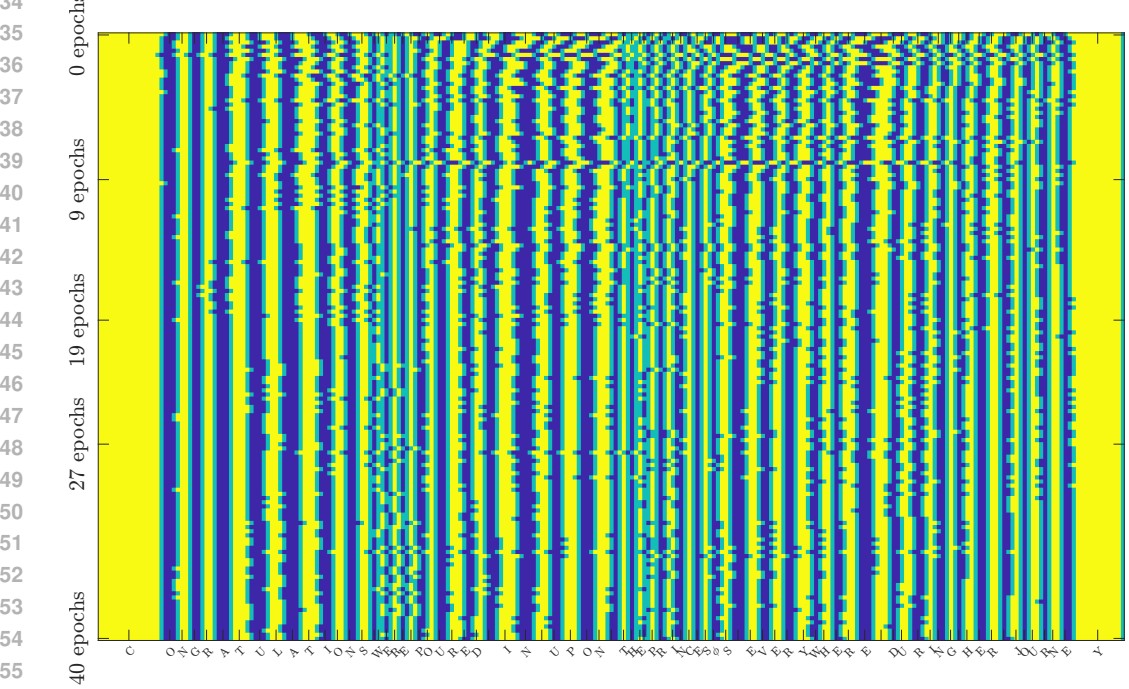

Figure 7: *Alignment evolution in the OTTC model during training for 40 epochs without freezing OT weights prediction head ($\alpha$ predictor).* On the $x$-axis, each pixel corresponds to one audio frame, while the $y$-axis represents the epoch. Frames grouped by tokens are shown in alternating colors (yellow and dark blue), with the boundaries of each group highlighted in light blue/green. One can note that during the initial phase of training, there is significant left/right movement of boundary frames for all groups. As training progresses, the movement typically stabilizes to around 1-2 frames.

of words with significant delay, or assigns very few audio frames to non-blank symbols, resulting in a peaky behavior.

**Temporal evolution of alignment.** An example of the evolution of the alignment in the OTTC model during training for 40 epochs without freezing *OT weights prediction head* is shown in Figure 7. Note that during the initial phase of training, there is significant left/right movement of boundary frames for all groups. As training progresses, the movement typically stabilizes to around 1-2 frames. While this can be considered "relatively stable" in terms of alignment, the classification loss (i.e., cross-entropy) in the OTTC framework is still considerably affected by these changes. This change of the loss is what impacts the final performance and the performance difference between freezing or not-freezing the alignments.

