# OpenReview forum: "OTTC: A differentiable alignment approach to automatic speech recognition"
_ICLR.cc/2025/Conference — Submitted to ICLR 2025_

### Official Review · Reviewer_6GZM · 2024-11-03

**Soundness:** 2
**Presentation:** 2
**Contribution:** 2
**Rating:** 3
**Confidence:** 3

**Summary:**

This paper aims to learn sequence to sequence prediction and alignment simultaneously. To achieve this, the authors define a pseud-metric called the Sequence Optimal Transport Distance (SOTD) over sequences based on one-dimensional optimal transport. SOTD enables the joint optimization of target sequence prediction and alignment.  They then derive the Optimal Temporal Transport Classification (OTTC) loss for automatic speech recognition (ASR). Experiments on the LibriSpeech and AMI datasets show that the proposed method achieves encouraging recognition accuracy, although it’s still worse than the popular sequence to sequence ASR modeling method Connectionist Temporal Classification (CTC). Besides, the alignment output from OTTC model does not have the peaky behavior observed in CTC-based models.

**Strengths:**

The optimal Temporal Transport Classification (OTTC) loss proposed in this paper has two parts, one is alignment related, based on Sequences Optimal Transport which is proved to be differentiable. Another is classification (or prediction) related, which is based on the cross-entropy loss. The advantage of this new loss is it enables the model to lean both the alignment and classification jointly. The authors explain and prove every mathematical theory behind this loss definition in detail. The authors also conduct proof-of-concept experiments on the ASR task and compare the results with CTC-based model. The results show the proposed method achieve reasonable speech recognition accuracy and alignment. Especially the alignment is not peaky as observed in CTC based ASR model.

**Weaknesses:**

The main weakness of the paper is that the experimental results didn’t show the better ASR accuracy of the proposed method when compared with CTC loss. It’s claimed that the alignment from the OTTC model is better than that from CTC model because it’s not peaky. But the author didn’t compare the alignment with the ground truth alignment to measure the alignment accuracy with convincing numbers. Below are some detailed comments:
•	Figure 4 and figure 5 show the alignment of OTTC model, but none of them show the “ground truth” alignment. And the paper didn’t present measurements of the alignment accuracy from OTTC model in other way. So, it doesn't seem credible to claim that OTTC could get better alignment since it’s not compared with the ground truth alignment.
•	The target of OT part of the loss is to find out optimal “alpha”(or alignment), it would be better if the author do some analyze of the value “alpha” of the trained model to show what’s the optimal value and does it have any physical meaning.
•	In this paper, the authors only show the results of uniform distribution of value “beta” and said learning the optimal beta is difficult. It would be better to show how the model will perform with other choice of beta (e.g. proportional to the letter duration) to show how “beta” will affect the results with different value? If the method is sensitive to the choice of “beta’. Then more work needs to be done to make this method applicable for real machine learning tasks.
Besides, there are some typos (or errors) in the paper. like:
•	In equation (3). If the dimension of gamma is n*m, and the dimensions of 1n is n*1. Then the multiplication of these two matrices is not valid. Similarly, the transpose of gamma has dimension m*n, it couldn’t be multiplied with the matrix with dimension m*1.
•	In equation (11). “A” in the left side of equal sign should be “W”, also “AW” in the right side of equal sign should be "W”.

**Questions:**

•	For OTTC model, the OT related parameters are frozen for the last 10 epochs in the experiment?  Why is number 10 used here and whether other values have been explored? Or how much will this parameter affect the results?
•	In section 6, it’s said that in the 960h-LibriSpeech training setup, it got 4.77% WER at epoch 30 and no meaningful improvement in WER at 40 epochs without freezing the OT weights. Does it mean the final WER is also around 4.77%? It’s also said the alignments remain relatively stable as training progresses. If so, freezing alignment vs. no freezing alignment shouldn’t have big difference, but based on table 1, freezing OT weights in the last 10 epoch could get 4.24% WER. Could the author explain more about this?

---

> ### Author Response · Authors · 2024-11-20
> **Author Response to Reviewer 6GZM (1/3)**
>
> We appreciate the reviewer’s valuable feedback and insightful comments. Below, we address each of the review points in detail.
>
> - **[W1]** *The main weakness of the paper is that the experimental results didn’t show the better ASR accuracy of the proposed method when compared with CTC loss.*
>
> Although the proposed OTTC loss currently lags behind CTC in terms of ASR performance, we would like to emphasize that the primary goal of this paper is to introduce a completely new framework for ASR. This framework offers end-to-end differentiability and addresses longstanding issues in existing approaches, such as the peaky behavior. The experiments serve mainly as proof of concept, demonstrating how the method works and its potential. With further work from the community, there is potential to develop a competitive and more cost-effective method based on this approach.
>
> While further testing the proposed OTTC framework after the paper submission deadline, we also experimented with different pre-trained models such as Wav2Vec2-large [1] instead of XLSR (as shown in the first submission). The obtained results depicted in the table below show that with such a pre-trained model, OTTC performance is very similar to that of CTC. These results further demonstrate the necessity to investigate the OTTC further and its potential of becoming a standard differentiable loss for ASR.
>
> | **Model** | **100h-LibriSpeech** |        | &#124; | **360h-LibriSpeech** |       | &#124; | **960h-LibriSpeech** |       |
> |-----------|----------------------|------------|------------|----------------------|------------|------------|----------------------|------------|
> |      | test-clean           | test-other | &#124;     | test-clean           | test-other | &#124;     | test-clean           | test-other |
> | **CTC**   | 3.36                 | 7.36       | &#124;     | 2.77                 | 6.58       | &#124;     | 2.20                  | 5.23       |
> | **OTTC**  | 3.77                 | 8.55       | &#124;     | 3.00                    | 7.44       | &#124;     | 2.52                 | 6.16       |
>
>
> [1] A. Baevski, Y. Zhou, A. Mohamed, and M. Auli, “wav2vec 2.0: A framework for self-supervised learning of speech representations,” in *NeurIPS,* 2020.
>
> - **[W2]** *It’s claimed that the alignment from the OTTC model is better than that from CTC model because it’s not peaky. But the author didn’t compare the alignment with the ground truth alignment to measure the alignment accuracy with convincing numbers. Below are some detailed comments: • Figure 4 and figure 5 show the alignment of OTTC model, but none of them show the “ground truth” alignment. And the paper didn’t present measurements of the alignment accuracy from OTTC model in other way. So, it doesn't seem credible to claim that OTTC could get better alignment since it’s not compared with the ground truth alignment.*
>
> Analyzing the alignment accuracy with concrete numbers is unfortunately not possible since one does not have access to the ground truth alignments (at least we are not aware of existing databases which also contain ground truth alignments). In ASR, we are only given the audio and the corresponding reference text, without a one-to-one alignment between the audio frames and text tokens. This is the reason why CTC and transducer models marginalize over all possible paths (alignments). In contrast, the proposed OTTC framework learns a single alignment, enabling end-to-end ASR while mitigating the peaky behavior commonly observed in CTC-based models.
>
> However, we agree with the reviewer that we should support our claim that the OTTC alignment avoids the peaky behavior observed in CTC, which is characterized by a significant proportion of audio frames being assigned to either the blank symbol or the space symbol (non-alphabet symbols) [2]. To this end, we calculated the average percentage of audio frames assigned to these two special symbols to quantitatively assess the model's peaky behavior. For the test-clean set, we found that **60.3%** of total frames in CTC models were assigned to these special symbols. In contrast, the OTTC model assigned only **22.9%** of frames to these symbols. This highlights the effectiveness of the alignment achieved by our proposed framework, which decisively avoids the extreme peaky behavior exhibited by CTC models.
>
> [2] Albert Zeyer, Ralf Schl ̈uter, and Hermann Ney, “Why does CTC result in peaky behavior?,” *arXiv preprint arXiv:2105.14849*, 2021.

---

> > ### Author Response · Authors · 2024-11-20
> > **Author Response to Reviewer 6GZM (2/3)**
> >
> > - **[W3]** *The target of OT part of the loss is to find out optimal “alpha”(or alignment), it would be better if the author do some analyze of the value “alpha” of the trained model to show what’s the optimal value and does it have any physical meaning.*
> >
> > Although it would be interesting to systematically derive a physical interpretation of an “alpha”, we are not aware of such an interpretation. Currently we hypothesize that the optimal alpha may denote precise localization of spoken letters in the audio, and its cumulative sum may signify the speaking rate, as this would likely be easier for the model's classification part to learn. However, additional analysis is necessary to finalize this relation.
> >
> > - **[W4]** *In this paper, the authors only show the results of uniform distribution of value “beta” and said learning the optimal beta is difficult. It would be better to show how the model will perform with other choice of beta (e.g. proportional to the letter duration) to show how “beta” will affect the results with different value? If the method is sensitive to the choice of “beta’. Then more work needs to be done to make this method applicable for real machine learning tasks.*
> >
> > We would like to thank the reviewer for bringing up this important point. Based on the reviewers’ recommendation, we conducted several experiments to show how the model would perform for other choices of beta.
> > 1) To show the importance of making beta learnable, we first experiment with learning beta using a trainable transformer decoder layer with tokenized reference text as input. We observe a degenerate solution in which all label weights (beta) are assigned to a single token, while all other tokens receive zero label weights, resulting in 100% WER. Intuitively, this behavior makes sense because the model can learn this shortcut, which still minimizes the loss (the loss goes to zero), as there are no constraints in the loss to prevent it. Next, we impose a constraint on the learnable beta values, ensuring they cannot fall below a certain threshold. However, we observe a slight degradation in performance, with around 1% degradation in WER for the 360-hour LibriSpeech setup.
> > 2) To further highlight the importance of beta, we conducted an oracle experiment.   In this experiment, we first force-align audio frames and text tokens using a CTC-based model trained on the same data. We then use this alignment to calculate beta values. In both 100h-LibriSpeech and 360h-LibriSpeech setups, the OTTC model matches the performance of the CTC model. This result underscores the critical role of the choice of beta (whether learnable or not). With an appropriate selection of beta, the OTTC model can achieve performance on par with, or potentially surpass, the CTC-based model.
> >
> > We will include these results in the revised version of the paper in Appendix A.3. Overall, the proposed framework shows great promise for ASR, and we hope our work paves the way for a new approach in the field
> >
> > - **[W5]** *Besides, there are some typos (or errors) in the paper. like: • In equation (3). If the dimension of gamma is nm, and the dimensions of 1n is n1. Then the multiplication of these two matrices is not valid. Similarly, the transpose of gamma has dimension mn, it couldn’t be multiplied with the matrix with dimension m1. • In equation (11). “A” in the left side of equal sign should be “W”, also “AW” in the right side of equal sign should be "W”.*
> >
> > All the typos have been addressed. Thank you for your feedback.
> >
> > - **[Q1]** *For OTTC model, the OT related parameters are frozen for the last 10 epochs in the experiment? Why is number 10 used here and whether other values have been explored? Or how much will this parameter affect the results?*
> >
> > In our investigations so far, we arbitrarily selected this number of epochs as a hyperparameter without any tuning. To further understand its impact, we conducted additional experiments on the 360h-LibriSpeech setup using the Wav2Vec2-large model, when freezing the OT weights prediction head for the last 5 and 15 epochs. When frozen for the last 5 epochs, we achieve a WER of 3.01; for the last 15 epochs, the WER is 3.10. As shown in the table above, freezing the OT head for the last 10 epochs results in WER of 3.00. Based on these results, it appears that the model’s performance doesn't change significantly given the model is trained for a few more epochs after freezing the alignment part of the OTTC model.

---

> > > ### Author Response · Authors · 2024-11-20
> > > **Author Response to Reviewer 6GZM (3/3)**
> > >
> > > - **[Q2]** *In section 6, it’s said that in the 960h-LibriSpeech training setup, it got 4.77% WER at epoch 30 and no meaningful improvement in WER at 40 epochs without freezing the OT weights. Does it mean the final WER is also around 4.77%? It’s also said the alignments remain relatively stable as training progresses. If so, freezing alignment vs. no freezing alignment shouldn’t have big difference, but based on table 1, freezing OT weights in the last 10 epoch could get 4.24% WER. Could the author explain more about this?*
> > >
> > > When alignments are not frozen for the last 10 epochs, the WER is around 4.77%. However, when alignments are frozen for the last 10 epochs, the WER we obtain (as reported in Table 1) is 4.24%.
> > >
> > > We believe that our use of the terms “alignments remain relatively stable” might not have been very precise and caused confusion. By stating that alignments remain relatively stable we meant that we experimentally observed changes only at the extremities of the consecutive group of frames assigned to a token. An example of evolution of alignment in the OTTC model during training for 40 epochs, without freezing OT weights prediction head (alpha predictor), is shown in Figure 7 (Appendix A.3.1) of the updated paper. Please note that during the initial phase of training, there is significant left/right movement of boundary frames for all groups. As training progresses, the movement typically stabilizes to around 1-2 frames.
> > >
> > > While this can be considered “relatively stable” in terms of alignment, the classification loss (i.e., cross-entropy) in the OTTC framework is still considerably affected by these changes. This change of the loss is what impacts the final performance and the difference between freezing or not-freezing the alignments. Based on the reviewers’ feedback, we have added this explanation in the revised version of the paper.
> > >
> > > We hope we have addressed the reviewer's concerns and would be happy to provide any additional details if needed. We sincerely hope the reviewer will consider revising the scores in light of the clarifications provided.

---

> > > ### Comment · Reviewer_6GZM · 2024-11-22
> > >
> > > * **[W4]**: These are interesting experiments, but they showed it's difficult to find optimal beta for this method now. This would be an important direction to improve this method.

---

> > ### Comment · Reviewer_6GZM · 2024-11-22
> >
> > Thanks for the detailed reply. below are my comments.
> > * **[W1]**:  The new results with Wav2Vec2-large  as the seed also shows the obvious gap between OTTC and CTC: > 10% relative WER increase. We couldn't say "OTTC performance is very similar to that of CTC" with such gap.
> > * **[W2]**:  Yes, there is no alignment information for the LibriSpeech set. But we could get alignment for it with some existing tools. like: https://github.com/pettarin/forced-alignment-tools. Avoiding peaky  behavior doesn't equal to improve alignment accuracy. I still think author should prove their claim " OTTC can learn accurate  alignment than CTC model " with proper metrics.

---

> ### Author Response · Authors · 2024-11-24
> **Author Response to Reviewer 6GZM**
>
> We thank the reviewer for their reply.
> - **[W1]** In presenting the new results using Wav2Vec2-large as the seed, our intent was to demonstrate the potential of our approach further, particularly as the gap with CTC is significantly reduced compared to the results obtained with XLSR in the original paper.
> We fully acknowledge the reviewer’s point that a relative WER increase of >10% shows that our approach is not yet on par with CTC. However, we would like to emphasize that our method introduces a completely novel framework, which is non-incremental in nature. We hope the community will recognize its potential and build upon it to advance its performance, much like how substantial research efforts were required to make CTC competitive with traditional hybrid systems (please see 2nd paragraph of Section 1 in [3]). We view this work as an important first step, and we believe it lays a foundation for further advancements.
>
> [3] D. Povey, V. Peddinti, D. Galvez, P. Ghahrmani, V. Manohar, X. Na, Y. Wang, and S. Khudanpur, “Purely sequence-trained neural networks for ASR based on lattice-free MMI,” in *INTERSPEECH,* 2016
>
> - **[W2]** We thank the reviewer for suggesting tools such as forced-alignment frameworks to obtain ground truth alignments. However, we initially refrained from using these tools due to the limitations of forced alignment as a substitute for true ground truth. During forced alignment generation, models predict probabilities over vocabularies (for CTC-based models) or PDF-IDs (for hybrid systems), and the Viterbi algorithm is subsequently used to infer frame-level alignments. This process is inherently influenced by the biases of the underlying models.
>
> Nevertheless, We identified that forced alignment from the AMI dataset has been used in past works [4, 5] to measure alignment performance. Following the methodology in [4], we calculated precision (P), recall (R), and F1 score. It is important to note that we only considered word-level timestamps, as they are typically less erroneous than individual phoneme or sub-word level timestamps. The results are as follows:
>
> CTC Model: P = 84.26, R = 83.62, F1 = 83.94
>
> OTTC Model: P = 84.77, R = 84.85, F1 = 84.81
>
> From these results, the OTTC model shows better alignment performance.
> However, this metric does not provide insight into the predicted duration of the words. To address this, we calculated an additional metric: Intersection Duration Ratio. This metric calculates the duration of overlap between the reference and predicted word segments, dividing it by the total reference duration of those words.
> The results for this metric are as follows:
>
> CTC Model: Intersection Duration Ratio = 17.19%
>
> OTTC Model: Intersection Duration Ratio = 42.12%
>
> This highlights that, on average, the CTC model either predicts the start of words with significant delay or assigns very few audio frames to non-blank symbols (resulting in a peaky behavior).
>
> [4] E. Rastorgueva, V. Lavrukhin, and B. Ginsburg, “NeMo Forced Aligner and its Application to Word Alignment for Subtitle Generation,” in *Proc. INTERSPEECH*, 2023.
>
> [5] Max Bain, Jaesung Huh, Tengda Han, and Andrew Zisserman. "WhisperX: Time-accurate speech transcription of longform audio," In *Interspeech*, 2023.
>
> - **[W4]** We thank the reviewer for their thoughtful feedback and for recognizing the value of our experiments. We agree that finding appropriate values for beta is an important direction for future research, which we believe will unlock the full potential of the proposed framework. This has been emphasized in the paper.
> However, we respectfully disagree with the statement that our experiments demonstrate finding an optimal beta is inherently difficult. The optimal beta may not necessarily be the one derived from forced-alignment using a CTC model, as such alignments can still suffer from issues like label delay. Instead, the oracle experiment serves to demonstrate that beta is a critical parameter, and further research is necessary to identify the truly optimal values.
>
> Even with the simple uniform beta used in our experiments, the proposed method achieves reasonable performance, showcasing the robustness and promise of the framework even in the absence of optimized beta values.

---

> > ### Author Response · Authors · 2024-11-25
> > **Author Response to Reviewer 6GZM**
> >
> > Dear Reviewer,
> >
> > We hope that our latest revisions have addressed your concerns. If there are any additional details or clarifications we can provide, please let us know, as the discussion period ends tomorrow (26th) AoE. Thank you for your time and valuable comments.
> >
> > Best,
> >
> > Authors

---

> > ### Comment · Reviewer_6GZM · 2024-11-27
> >
> > I agree that the proposed method introduces a completely novel framework to find the alignment between two sequences other than CTC. But my concern is still the lower accuracy. Besides, how to tune parameter "beta" need more investigations.

---

> > > ### Author Response · Authors · 2024-11-29
> > > **Author Response to Reviewer 6GZM**
> > >
> > > While we acknowledge the performance gap with CTC, as you noted, we believe our work marks a significant first step in introducing a differentiable alignment framework with the flexibility to define distances/metrics over sequences (e.g., SOTD in our paper). Our experiments show that the framework has potential, and with further work from the community, it may eventually surpass state-of-the-art results.
> > >
> > > We have now uploaded a revised version of our manuscript, incorporating experiments and insights based on the discussion with you and other reviewers (please refer to our global comment for details). We kindly invite you to review the changes, and we hope you will reconsider your scores in light of these updates.

---

### Official Review · Reviewer_wARR · 2024-11-03

**Soundness:** 3
**Presentation:** 4
**Contribution:** 2
**Rating:** 6
**Confidence:** 4

**Summary:**

The authors propose a noval end-to-end loss for automatic speech recognition (ASR), called Optimal Temporal Transport Classification (OTTC), jointly learning temporal alignment and audio frame classification. This loss is derived from the introduced Sequence Optimal Transport Distance (SOTD) framework, which constructs pseudo-metrics over the sequence space. Central to this framework is a parameterized and differentiable alignment model based on one-dimensional optimal transport, offering linear complexity in both time and space. Experimental results on Librispeech and AMI datasets demonstrate that the proposed method achieves promising performance.

**Strengths:**

- The idea of this work is novel and non-incremental, supported by detailed mathematical theoretical proofs. Meanwhile, the writing style is excellent and impressive.
- In theory, the proposed method achieves linear complexity both in time and space. Experimental results demonstrate that it mitigates the peaky behavior observed in the Connectionist Temporal Classification (CTC) models.

**Weaknesses:**

- The performance of the proposed method is noticeably inferior to that of CTC, which significantly restricts its applicability in real-world scenarios, especially considering that CTC already lags behind transducer and hybrid systems combining CTC and attention-decoder.
- The authors don't provide ablation experiments for the proposed OTTC model. I would suggest at least testing removing the OT weight prediction head and using fixed and uniform OT weights instead.

**Questions:**

- While OTTC theoretically offers linear complexity both in time and space, how about the practical training cost in terms of the GPU memory usage and training time compared to CTC?
- In line 520-521, the authors state, "Furthermore, our framework effectively addresses the peaky behavior commonly seen in CTC models, resulting in improved alignments". To validate the claim of improved alignments, I would suggest computing quantitative metrics by comparing the decoding timestamps and the pre-computed ground-truth token-time alignments.
- I noticed several possible typos:
  - In line 156-157, "alignement" should be corrected to "alignment".
  - In line 253-254, "[" should be corrected to "]".
  - In Equation 3, should "$\gamma 1_n=\alpha$ and $\gamma^T 1_m=\beta$" be "$\gamma 1_m=\alpha$ and $\gamma^T 1_n=\beta$"?
  - In Equation 11, should "A" be corrected to "W"?
  - In Equation 12, should "$\log p_{l_j} (x_i)$" be corrected to "$\log p_{l_{y_j}} (x_i)$"?

---

> ### Author Response · Authors · 2024-11-20
> **Author Response to Reviewer wARR**
>
> We sincerely thank the reviewer for their valuable feedback and insightful comments, which have helped us improve the clarity and quality of our work. Below, we address each of the review points in detail.
>
> **[W1]** Most recent losses in ASR are variations and adaptations of CTC, including the transducers mentioned by the reviewer. This paper, however, was primarily intended to present a new proposal for accomplishing the ASR task entirely independent of CTC, enabling the community to test and build on this new idea. The experiments serve mainly as proof of concept, demonstrating how the method works and its potential. With further work from the community, there is potential to develop a competitive and more cost-effective method based on this approach.
>
> While further testing the proposed OTTC framework after the paper submission deadline, we also experimented with different pre-trained models such as Wav2Vec2-large [1] instead of XLSR (as shown in the first submission). The obtained results depicted in the table below show that with such a pre-trained model, OTTC performance is very similar to that of CTC. These results further demonstrate the necessity to investigate the OTTC further and its potential of becoming a standard differentiable loss for ASR.
>
> | **Model** | **100h-LibriSpeech** |        | &#124; | **360h-LibriSpeech** |       | &#124; | **960h-LibriSpeech** |       |
> |-----------|----------------------|------------|------------|----------------------|------------|------------|----------------------|------------|
> |      | test-clean           | test-other | &#124;     | test-clean           | test-other | &#124;     | test-clean           | test-other |
> | **CTC**   | 3.36                 | 7.36       | &#124;     | 2.77                 | 6.58       | &#124;     | 2.20                  | 5.23       |
> | **OTTC**  | 3.77                 | 8.55       | &#124;     | 3.00                    | 7.44       | &#124;     | 2.52                 | 6.16       |
>
>
> [1] A. Baevski, Y. Zhou, A. Mohamed, and M. Auli, “wav2vec 2.0: A framework for self-supervised learning of speech representations,” in *NeurIPS,* 2020.
>
> **[W2]** Removing the weights prediction is equivalent to not searching for the best path, but instead maximizing the probability of the path where a frame is assigned to the same label during the course of training. In this case, we completely lose the localization of the letter in the audio, which is why we didn’t  include such experiments in the paper.
> Following the reviewer’s feedback, we conducted an experiment using the 360-hour LibriSpeech setup with Wav2Vec2-large as the pre-trained model, employing fixed and uniform OT weights. We observed a WER of 3.51 on test-clean (compared to 2.77 for CTC and 3.0 for OTTC with learnable OT weights) and 8.24 on test-other (compared to 6.58 for CTC and 7.44 for OTTC with learnable OT weights). From these results, we note that while the WERs are slightly degraded in comparison to OTTC with learnable OT weights, the localization is completely lost.
>
> **[Q1]** While we agree with the reviewer that the practical training cost is an important characteristic to analyze, unfortunately we cannot provide a fair comparison at the moment. In our current implementation, OTTC is slightly slower to train than CTC (approximately 60 samples per seconds vs 90 samples per seconds for CTC), because CTC is directly implemented and optimized  in C++, while OT uses vectorisation from the torch library. However,  when appropriately optimized, OT is expected to be faster than CTC.
>
> **[Q2]** The peaky behavior of CTC models is characterized by a significant proportion of audio frames being assigned to either the blank symbol or the space symbol (non-alphabet symbols) [2]. Following the reviewer’s feedback, we calculated the average percentage of audio frames assigned to these two special symbols to quantitatively assess the model's peaky behavior. For the test-clean set, we found that 60.3% of total frames in CTC models were assigned to these special symbols. In contrast, the OTTC model assigned only 22.9% of frames to these symbols. This highlights the effectiveness of the alignment achieved by our proposed framework, which decisively avoids the extreme peaky behavior exhibited by CTC models.
>
> While it would be indeed important to further validate our claim using ground-truth toke-time alignments, to the best of our knowledge, no databases with such ground-truth information are currently available.
>
> [2] Albert Zeyer, Ralf Schl ̈uter, and Hermann Ney, “Why does CTC result in peaky behavior?,” *arXiv preprint arXiv:2105.14849*, 2021.
>
> **[Q3]** We appreciate the reviewer for highlighting the typos. We have addressed them, and the corrections will be reflected in the revised version of our paper.

---

> > ### Comment · Reviewer_wARR · 2024-11-25
> > **Thanks for the detailed reply!**
> >
> > Thanks for the detailed reply! I still have a few concerns:
> >
> > 1. I understand that removing the weights prediction could lead to a loss in localization, but I'm curious if the model could still be trained successfully without it? (Just a question for discussion, I don't require the need of an experiment here.)
> >
> > 2. The performance gap between CTC and OTTC is still noticeable, even with Wav2Vec2-large, which is pretrained: (e.g., 2.20/5.23 vs. 2.52/6.16 with 960h-LibriSpeech) This raises concerns about the practical applicability of this method. Is there room for improvement?"

---

> ### Author Response · Authors · 2024-11-25
> **Author Response to Reviewer wARR**
>
> We thank the reviewer for their reply.
>
> **1.** Yes, the model can still be trained successfully without the OT weights prediction. As demonstrated in our previous experiment (detailed in the response above), we trained the model on the 360-hour LibriSpeech setup with Wav2Vec2-large as the pre-trained model, using fixed and uniform OT weights. The results showed a WER of 3.51 on test-clean (compared to 2.77 for CTC and 3.0 for OTTC with learnable OT weights) and 8.24 on test-other (compared to 6.58 for CTC and 7.44 for OTTC with learnable OT weights).
>
> These results indicate that while the model can indeed be trained, the use of fixed OT weights leads to degraded WERs and a complete loss of localization. We hypothesize that fixed OT weights ($\boldsymbol{\alpha}$) struggle to account for variations in speaking rates across the training set, which may explain the observed challenges in performance.
>
> **2.** We thank the reviewer for bringing up this important point. To further investigate the potential of the OTTC framework, we conducted an oracle experiment. In this experiment, we used a CTC-based model trained on the same dataset to force-align audio frames and text tokens, thereby generating oracle $\boldsymbol{\beta}$ values. These $\boldsymbol{\beta}$ values were then used to train the OTTC model. Remarkably, under this setup, the OTTC model matched the performance of the CTC model on both 100h-LibriSpeech and 360h-LibriSpeech setups.
> These results underscore the promise of OTTC as a framework for ASR. With an appropriately chosen fixed or learnable $\boldsymbol{\beta}$, OTTC can achieve performance on par with, or potentially surpass, that of CTC, while avoiding the peaky behavior characteristic of CTC.
>
> We hope we have addressed the reviewer's concerns and would be happy to provide any additional details if needed. We sincerely hope the reviewer will consider revising the scores in light of the clarifications provided.

---

> > ### Comment · Reviewer_wARR · 2024-11-25
> > **Thanks for the detailed reply!**
> >
> > Thanks for reply.
> >
> > I would suggest including above experimental results (removing the OT weight prediction head and using fixed and uniform OT weights) in the revised manuscript.
> >
> > By the way, to obtain the token-time alignments, you could use toolkits such as:
> > - Montreal Forced Aligner, https://montreal-forced-aligner.readthedocs.io/en/v3.1.2/index.html#
> > - torchaudio: https://pytorch.org/audio/stable/tutorials/forced_alignment_tutorial.html
> >
> > I would also suggest including quantitative metrics to show the proposed method can get improved alignments.

---

> > > ### Author Response · Authors · 2024-11-25
> > > **Author Response to Reviewer wARR**
> > >
> > > We thank the reviewer for their reply.
> > >
> > > Thanks for the suggestion. We will include these details and experimental results in the revised manuscript.
> > >
> > > We thank the reviewer for suggesting tools such as forced-alignment frameworks to obtain ground truth alignments. However, we initially refrained from using these tools due to the limitations of forced alignment as a substitute for true ground truth. During forced alignment generation, models predict probabilities over vocabularies (for CTC-based models) or PDF-IDs (for hybrid systems), and the Viterbi algorithm is subsequently used to infer frame-level alignments. This process is inherently influenced by the biases of the underlying models.
> > >
> > > Nevertheless, We identified that forced alignment from the AMI dataset has been used in past works [3, 4] to measure alignment performance. Following the methodology in [3], we calculated precision (P), recall (R), and F1 score. It is important to note that we only considered word-level timestamps, as they are typically less erroneous than individual phoneme or sub-word level timestamps. The results are as follows:
> > >
> > > CTC Model: P = 84.26, R = 83.62, F1 = 83.94
> > >
> > > OTTC Model: P = 84.77, R = 84.85, F1 = 84.81
> > >
> > > From these results, the OTTC model shows better alignment performance.
> > > However, this metric does not provide insight into the predicted duration of the words. To address this, we calculated an additional metric: Intersection Duration Ratio. This metric calculates the duration of overlap between the reference and predicted word segments, dividing it by the total reference duration of those words.
> > > The results for this metric are as follows:
> > >
> > > CTC Model: Intersection Duration Ratio = 17.19%
> > >
> > > OTTC Model: Intersection Duration Ratio = 42.12%
> > >
> > > This highlights that, on average, the CTC model either predicts the start of words with significant delay or assigns very few audio frames to non-blank symbols (resulting in a peaky behavior).
> > >
> > > [3] E. Rastorgueva, V. Lavrukhin, and B. Ginsburg, “NeMo Forced Aligner and its Application to Word Alignment for Subtitle Generation,” in *Proc. INTERSPEECH,* 2023.
> > >
> > > [4] Max Bain, Jaesung Huh, Tengda Han, and Andrew Zisserman. WhisperX: Time-accurate speech transcription of longform audio. In *Interspeech,* 2023.
> > >
> > > We will include the performance of these alignment metrics in the revised manuscript as well.

---

> > > > ### Comment · Reviewer_wARR · 2024-11-25
> > > > **Thanks for the detailed reply!**
> > > >
> > > > OK. I have updated the score from 5 to 6.

---

### Official Review · Reviewer_bmks · 2024-11-04

**Soundness:** 3
**Presentation:** 4
**Contribution:** 3
**Rating:** 8
**Confidence:** 4

**Summary:**

This is a well-written submission describing a novel application of Optimal Transport to the fundamental alignment problem in ASR. Re-casting alignments as a matrix of coupling weights representing the "transport" of units in a source alignment to units in a target alignment the authors to use concepts from Optimal Transport to minimize the overall transport cost efficiently and flexibly, in a way that mitigates the peaky behavior typically observed in ASR alignments based on the CTC model with dynamic programming. The work presents ASR results (WERs) on well-known public domain tasks (LibriSpeech and AMI); the proposed method trails the standard CTC model, but the work offers a fresh perspective on a long-standing challenge in core ASR modeling technologies. As such I think the work is a high interest to the community.

**Strengths:**

Clarity of presentation, quality of the writing, and originality; very good literature survey & references.

**Weaknesses:**

A few concepts could be explained more clearly. One thought regarding the evaluation: since the proposed method only learns a single alignment path, it might make sense to include a comparison with CTC models trained using the single best alignment paths for any given training utterance (aka the "Viterbi algorithm") rather than the standard sum over all possible alignments. I am wondering if the gap in WER between proposed method and standard CTC comes from the use of a single path, versus multiple paths. This could be part of the evaluation.

**Questions:**

Adding to my earlier comment, if indeed a limitation of the work is that only a single path is learned, and though this of practical/computational interest, if further investigation were to reveal that this actually hurts generalization, could one formulate SOTD so as to learn multiple transports for each utterance?

More comments, mostly nits re: writing:

L048, "requires comparatively large amount of data" --> "requires a comparatively large amount of data"

L128: re: the "d-dimensional vector sequences": the writing suggests that x_i and y_i are both d-dimensional, is that intended...?

L190: specify what is a "δ measure"?

L195, "ν[β, n]", I think this should be "ν[β, m]"?

L207, " coupling matrix γ∗ ": is there a more informative term? It suggests this is an alignment matrix, but then AIU each entry is actually an amount of mass moved from i to j.

L223, "α→γ∗ =argmin_{γ∈Γ} W(μ[α,n],ν[β,m])," give the reader a heads up, How will γ typically be found... gradient descent, or some other method? Give an intuition about the east/difficultly therein?

L241, "The computational cost of these alignment functions is low," explain why? (relating to previous comment). (The cost seems to go beyond just the bins being sorted or not, but perhaps that is in fact the key aspect it is not completely clear to me).

L252, "Sequences Optimal Transport Distance (SOTD)": motivate this extension more? I.e., make it clear what is missing from the alignment model presented so far.

L271, "there is sequences" : fix typo

L302, "When the function F is powerful, the model can collapse ": be more precise than "powerful"? What types of specific functions would lead to collapse?

L325, "Ce", define Cross-Entropy somewhere? This would make e.g. Eq. (14) clearer .

L376, "relaxation of the last term ": what does "relaxation" here and in the following mean...?

L455, "peror-mance", fix typo

L520, "envision that learning label weights with suitable constraints can bridge the performance gap", be more specific?

L845, "SOTD ARE PSEUDO METRIC" --> "SOTD IS A PSEUDO METRIC"?

---

> ### Author Response · Authors · 2024-11-20
> **Author Response to Reviewer bmks (1/2)**
>
> We sincerely thank the reviewer for their valuable feedback and insightful comments, which have helped us improve the clarity and quality of our work. Below, we address each of the review points in detail.
>
> **[W1]** Thank you for the suggestion. Based on the recommendation, we conducted an experiment where we first obtained the best path (forced-alignment using the Viterbi algorithm) from a trained CTC-based model on the same dataset and then trained a model to learn this single best path using Cross-Entropy. On the 360-hour LibriSpeech setup with Wav2Vec2-large [1] as the pre-trained model, this approach achieved a WER of 7.04 on the test-clean set and 13.03 on the test-other set. In comparison, the OTTC model under the same setup achieved significantly better results, with a WER of 3.00 on test-clean and 7.44 on test-other. These findings suggest that future work on single-path alignment strategies could benefit from building upon our proposed framework.
>
> [1] A. Baevski, Y. Zhou, A. Mohamed, and M. Auli, “wav2vec 2.0: A framework for self-supervised learning of speech representations,” in *NeurIPS,* 2020.
>
> - **[Q1]** *Adding to my earlier comment, if indeed a limitation of the work is that only a single path is learned, and though this of practical/computational interest, if further investigation were to reveal that this actually hurts generalization, could one formulate SOTD so as to learn multiple transports for each utterance?*
>
> In our earlier investigations, we explored learning multiple heads for both logits and OT weights ($\boldsymbol{\alpha}$). The outputs from the OT prediction heads were averaged using either the geometric or arithmetic mean for alpha, and the arithmetic mean for logits.
> Although this approach led to faster convergence, it ultimately resulted in worse performance, prompting us to abandon this direction.
>
> We believe the best way to match or surpass the state of the art is by making $\boldsymbol{\beta}$ learnable with appropriate constraints or by refining the selection of static $\boldsymbol{\beta}$. Towards this end, we conducted an oracle experiment. In this experiment, we first force-align audio frames and text tokens using a CTC-based model trained on the same data. We then use this alignment to calculate $\boldsymbol{\beta}$ values. For example, given the target sentence "YES" and the best valid path from the Viterbi algorithm was "($\phi Y \phi \phi E E S$)" we re-labeled it to "($\phi Y \phi E S$)" and set $\boldsymbol{\beta}$ = [1/7, 1/7, 2/7, 2/7, 1/7].
> This approach enabled OTTC to learn a uniform distribution for $\boldsymbol{\alpha}$, mimicking CTC's highest probability path. As a result, in both the 100h-LibriSpeech and 360h-LibriSpeech setups, the OTTC model converged much faster and matched CTC performance. This experiment underscores the critical role of $\boldsymbol{\beta}$, suggesting that a better strategy for its selection or training could lead to further improvements.
>
> - **[Q2]** *L128: re: the "d-dimensional vector sequences": the writing suggests that x_i and y_i are both d-dimensional, is that intended...?*
>
> Yes, this is intentional. In the ASR context, $x_i$ represents a frame, while $y_j$ is the label, typically encoded as a one-hot vector. More broadly, the SOTD framework can be applied to any type of data (e.g., vectors, matrices, or tensors), provided a differentiable distance can be computed in the respective space.
>
> - **[Q3]** *L207, " coupling matrix γ∗ ": is there a more informative term? It suggests this is an alignment matrix, but then AIU each entry is actually an amount of mass moved from i to j.*
>
> The term "coupling matrix" for $\boldsymbol{\gamma}$ is commonly used in optimal transport (OT). While it could potentially be confused with an alignment, which represents a discrete binary relation between bins (either connected or not), we opted not to refer to $\boldsymbol{\gamma}$ as a "soft coupling" to avoid any confusion for readers familiar with OT theory. Although $\boldsymbol{\gamma}$ describes the continuous relationship between bins in terms of the mass moved, we chose to retain the term "coupling" and referred to the usual alignment as a "discrete alignment" for clarity.
>
> - **[Q4]** *L223, "α→γ∗ =argmin_{γ∈Γ} W(μ[α,n],ν[β,m])," .....* and *L241, "The computational cost of these alignment functions is low," explain why? (relating to previous comment).....*
>
> The monotonicity described above is the key property for computing these quantities. If the bins are not sorted, the complexity is O(max(n, m) log(max(n, m))); otherwise, it is O(max(n,m)). An intuitive algorithm for this computation is provided in Appendix 1.1.1. In response to the reviewer’s question, we have expanded this appendix to further clarify both the computation and complexity. Additionally, we have included a figure in this appendix to visually illustrate the process.

---

> ### Author Response · Authors · 2024-11-20
> **Author Response to Reviewer bmks (2/2)**
>
> - **[Q5]** *L252, "Sequences Optimal Transport Distance (SOTD)": motivate this extension more? I.e., make it clear what is missing from the alignment model presented so far.*
>
> The alignment introduced so far is a differentiable function that maps vectors $\boldsymbol{\alpha}$ and $\boldsymbol{\beta}$ to an alignment. This serves as a general alignment operation.
> SOTD utilizes this alignment to compute distance measures between sequences, which necessitates the introduction of a cost function. Specifically, SOTD represents one approach to comparing sequences using this function, but other methods could also be developed based on this alignment.
>
> The contribution is therefore two-fold: the introduction of this differentiable alignment function and the development of SOTD, which employs it to define a distance measure between sequences.
>
> - **[Q6]** *L302, "When the function F is powerful, the model can collapse ": be more precise than "powerful"? What types of specific functions would lead to collapse?*
>
> By 'powerful,' we mean a function capable of adapting and performing a wide range of transformations. In soft-DTW, only the first and last elements of sequences are guaranteed to align, while all in-between frames or targets may be ignored; i.e., there is no guarantee that soft-DTW will yield a discrete monotonic alignment. A 'powerful' transformation $F$ can map $\mathbf{x}$ to $F(${$\mathbf{x}$}$)$ in such a way that soft-DTW ignores the in-between transformed frames ($F(${$\mathbf{x}$}$)$) and targets ({$\mathbf{y}$}), which we refer to as a collapse. This is why transformations learned through sequence comparison are typically constrained (e.g., to geometric transformations like rotations) [2]. Transformer architectures, being powerful, are prone to collapse, as demonstrated in a new experiment we conducted using soft-DTW as a loss function. On the 360h-LibriSpeech setup with Wav2Vec2-large [1] model, the best WER achieved using soft-DTW is 39.43. In comparison CTC yields 2.77, and our proposed OTTC yields 3.00. One of the key advantages of our method is that, by construction, such a collapse is not possible.
> Based on the reviewer’s feedback, we will include these explanations and results in the revised version of the paper.
>
> [2] Titouan Vayer, L. Chapel, N. Courty, Rémi Flamary, Yann Soullard, and R. Tavenard, “Time series alignment with global invariances,” *ArXiv, abs/2002.03848,* 2020a
>
> - **[Q7]** *L376, "relaxation of the last term ": what does "relaxation" here and in the following mean...?*
>
> In mathematical optimization, continuous relaxation refers to interpreting a discrete problem in a continuous manner. CTC paths represent discrete alignments, which are non-differentiable by nature. SOTD is a relaxation of a single path, as the alignment is now represented by $\boldsymbol{\gamma}$, which is continuously valued and differentiable. This approach ensures the differentiability of the path. For clarity, we have explicitly stated in the revised version of the paper that this refers to continuous relaxation.
>
> - **[Q8]** *L520, "envision that learning label weights with suitable constraints can bridge the performance gap", be more specific?*
>
> As mentioned earlier, we conducted an oracle experiment where the label weights ($\boldsymbol{\beta}$) were derived from CTC, and observed that OTTC matches CTC performance in this case. Thus far, we have not found a more effective approach for selecting $\boldsymbol{\beta}$ than using a uniform distribution. While making \$\boldsymbol{\beta}$ trainable is an option; however, if it is unconstrained, the model risks collapsing as seen with soft-DTW, as there would be no guarantee that every target is reached.
> The goal is to make $\boldsymbol{\beta}$ trainable while ensuring it does not contain zero coefficients, without excessively constraining it. This remains a challenging task, and our attempts so far have yielded only limited success. We will continue our investigations and hope the broader community may uncover effective solutions to this problem.
>
> - **[Q9]** *L048...., L190...., L195...., L271...., L325...., L455....,* and *L845....*
>
> We thank the reviewer for carefully reading our paper and appreciate highlighting the typos. We have addressed them in the revised version of the paper.

---

> > ### Comment · Reviewer_bmks · 2024-12-03
> >
> > Thank you for your responses, and for incorporating some of my feedback. I think the paper is (even) stronger now, but I will keep my original score. This is definitely an interesting way of framing the alignment problem -- the challenge now is to beat the state of the art.

---

### Author Response · Authors · 2024-11-29
**Revised version of the paper**

Dear Reviewers,

Thank you for your valuable feedback. In response, we have added a new section (A.3) in the appendix, which includes the following updates:

- **Expanded Results:** We evaluate the OTTC framework using Wav2Vec2-large as the pre-trained model, further narrowing the performance gap between OTTC and CTC.
- **Ablation Studies:** We conduct experiments on key design choices, such as single-path alignment, fixed OT weights, and the impact of freezing the *OT weights prediction head* across epochs. We also highlight the importance of learnable $\boldsymbol{\beta}$ and include an oracle experiment showcasing OTTC’s potential to match or potentially exceed CTC. Additionally, we discuss the limitations of soft-DTW, such as its susceptibility to alignment collapse, and provide a comparative analysis with OTTC.
- **Alignment Analysis:** We provide quantitative metrics (precision, recall, F1 score, intersection duration ratio) and visualizations to evaluate alignment performance. Our results show that OTTC achieves better alignment performance compared to CTC. Additionally, we discuss the temporal evolution of alignments during training.

We hope these additions comprehensively address your suggestions and improve the clarity of our work.

Best,

The Authors

---

### Meta-Review · Area_Chair_Lb1w · 2024-12-17

**Metareview:**

The authors introduce a novel end-to-end loss for automatic speech recognition (ASR) called Optimal Temporal Transport Classification (OTTC), which simultaneously learns temporal alignment and audio frame classification. This loss is derived from the proposed Sequence Optimal Transport Distance (SOTD) framework, which establishes pseudo-metrics over the sequence space. A key component of this framework is a parameterized, differentiable alignment model based on one-dimensional optimal transport, providing linear complexity in both time and space.

 The key strength of the work is its originality, since the technique question address the CTC "peaky" behavior is novel as indicated by the three independent reviewers. The key concern is the performance, which is behind both CTC and state-of-the-art. Only after the reviewers' requests for additional experiments the proposed technique after some twists and tricks reduces the gap with CTC.  Furthermore,  the tune parameter "beta" needs more investigations.

Overall, it should be the authors' responsibility to demonstrate that the proposed approach can outperform the state-of-the-art within the submitted work, rather than leaving this to future research.

**Additional Comments On Reviewer Discussion:**

The paper is well-written, which made the review process much easier for the reviewers. The discussion phase was engaging, with a good collaboration between the authors and the reviewers. All reviewers acknowledged the originality of the work, but they also noted issues with its current performance (lagging behind SOTA). Reviewer bmks placed greater emphasis on the originality, recommending the paper for acceptance. In contrast, Reviewer 6GZM focused more on the performance gap with the current state-of-the-art. Lastly, Reviewer wARR increased their rating during the discussion phase, but the score was increased only to 6  due to concerns about the current model's performance. It should be noted that even the most positive reviewer concluded their final response to the authors by stating, "the challenge now is to beat the state of the art.

---

### Decision · Program_Chairs · 2025-01-22

Reject